# Embedding-Aligned Language Models

**Guy Tennenholtz**[†][*], **Yinlam Chow**[‡], **Chih-Wei Hsu**[†], **Lior Shani**[†], **Ethan Liang**[‡], **Craig Boutilier**[†]

† Google Research, ‡ Google Deepmind

## Abstract

We propose a novel approach for training large language models (LLMs) to adhere to objectives defined within a latent embedding space. Our method leverages reinforcement learning (RL), treating a pre-trained LLM as an environment. Our *embedding-aligned guided language* (EAGLE) agent is trained to iteratively steer the LLM's generation towards optimal regions of the latent embedding space, w.r.t. some predefined criterion. We demonstrate the effectiveness of the EAGLE agent using the MovieLens 25M and Amazon Review datasets to surface content gaps that satisfy latent user demand. We also demonstrate the benefit of using an optimal design of a state-dependent action set to improve EAGLE's efficiency. Our work paves the way for controlled and grounded text generation using LLMs, ensuring consistency with domain-specific knowledge and data representations.

## 1   Introduction

Large language models (LLMs) such as Gemini [Team et al., 2023] and GPT [Achiam et al., 2023] have revolutionized the field of natural language processing, achieving remarkable success in text generation, translation, comprehension, as well as expert-level performance on challenging tasks (e.g., exams, coding). However, effectively applying (or fine-tuning) LLMs to *domain-specific* tasks often requires considerable domain knowledge and labeled human data [Jeong et al., 2023, Ouyang et al., 2022, Ziegler et al., 2019]. Fortunately, in many cases, domain knowledge is already captured and encoded within *latent embedding spaces*.

Latent embeddings are continuous vectors, ubiquitous in fields such as recommender systems [Hansen et al., 2020], reinforcement learning (RL) [Nabati et al., 2023, Pertsch et al., 2021], and image classification [Girdhar et al., 2023, Radford et al., 2021]. They offer a powerful means to represent entities, concepts, and relationships within a specific domain. For example, in recommender systems, embeddings of items and users encapsulate information about preferences and behavior [Zhao et al., 2023], embeddings of images can capture their content and style [Radford et al., 2021], while embeddings of scientific articles can represent their research area and findings [Taher Harikandeh et al., 2023]. The utility of a latent embedding lies in its underlying compact representation of entities, concepts or relationships, and the associated metrics, allowing one to construct simpler, more efficient models or induce control over various processes [Arvanitidis et al., 2018, 2021, Radford et al., 2021, Tennenholtz and Mannor, 2022]. Importantly, latent embeddings are often pre-computed, and can thus serve as a readily available rich source of domain knowledge.

This leads to the key question: **can we leverage latent embeddings to better control and guide LLM generation?** In this paper, we present a novel framework which accomplishes this by exploiting latent embedding spaces to define an objective function for an LLM in an iterative RL-driven process.

As an example, consider the challenge of assisting content creators in generating valuable content within a recommender ecosystem (e.g., YouTube, Reddit, Spotify) [Boutilier et al., 2024]. An

---

[*]Correspondence to: `guytenn@gmail.com`

important aspect of this problem includes identifying and surfacing of *content gaps*, i.e., identifying hypothetical content which could potentially drive value for users, and subsequently, describing it to creators. Latent embeddings offer an effective way of defining a content gap. Informally, a content gap is a hypothetical (i.e., non-existing) content item, corresponding to some point in a latent embedding space for which: (1) no content currently exists, implying an unexplored area within the existing content landscape, and (2) adding this content would improve overall system welfare (i.e., drive net positive value for both users and creators). While the latent embedding representation facilitates identification of content gaps, describing or *surfacing* this hypothetical continuous vector to its potential creator requires that we move beyond the latent representation. The content gap embedding must be translated into a description that communicates its essence to a creator in a clear and actionable manner, requiring some way of interpreting the latent embedding.

In this paper, we address the general problem of *aligning LLMs with latent embedding spaces*. We address this by working directly with both the embedding space and the generated outputs of an LLM. We formulate an RL-driven framework which iteratively steers an LLM towards regions of an embedding space deemed optimal according to some predefined criterion (e.g., a content gap). To do so, we use a language-based agent to guide the LLM by modifying a textual representation of an entity. We then embed the resulting description into the latent embedding space to assess its quality w.r.t. our predefined criterion. Finally, we use this feedback to guide the LLM's next modification, steering it closer to optimal regions of the latent space.

Our contributions are as follows: (1) We propose an Embedding-Aligned Guided LanguagE (EAGLE) agent, which aligns an LLM with a latent embedding space; (2) We provide a comprehensive framework for novel content creation using LLMs, formulating a new technique for designing exploratory and high quality action sets by leveraging the power of LLMs, latent embeddings, and *G-optimal design* [Zhu et al., 2022]; (3) We validate the effectiveness of our approach on the MovieLens 25M dataset [Harper and Konstan, 2015], aligning creation to behavioral movie and user embeddings. Our results are evaluated by human raters demonstrating EAGLE's ability to guide an LLM to generate text that is both creative and consistent with the domain's data representations.[2]

## 2 Preliminaries and Related Work

**Large Language Models (LLMs).** An LLM $\mathcal{L} : \mathcal{S} \mapsto \Delta_{\mathcal{S}}$ maps sequence of tokens to probabilities over sequences of tokens. Transformers [Vaswani et al., 2017] are often used to train such LLMs over vast amounts of data, predicting the next token $x_t$ in a sequence given the preceding tokens $(x_1, x_2, \ldots, x_{t-1})$, by minimizing the cross-entropy loss. Pre-trained LLMs often vary in size, with larger models having greater capabilities [Achiam et al., 2023, Team et al., 2023, Touvron et al., 2023]; e.g., Llama 2 [Touvron et al., 2023] models have 7B, 13B, and 70B parameter variants.

**Embedding Language Models (ELMs).** An *embedding language model (ELM)* [Tennenholtz et al., 2024] combines textual prompts with vector representations of domain knowledge. It uses adapter layers to connect embedding inputs to a text-only pre-trained LLM. By training the model on these combined inputs, ELM generates language that reflects the information contained in the domain embedding vectors, even for hypothetical entities [Tennenholtz et al., 2024].

**Reinforcement Learning (RL).** A *Markov decision process (MDP)* is a tuple $(\mathcal{S}, \mathcal{A}, P, r, H, \gamma)$, where $\mathcal{S}$ is the state space, $\mathcal{A}$ is the action space, $P : \mathcal{S} \times \mathcal{A} \mapsto \Delta_{\mathcal{S}}$ is the transition function, $r : \mathcal{S} \times \mathcal{A} \mapsto \mathbb{R}$ is the reward function, $H$ is the horizon, and $\gamma \in [0, 1]$ is the discount factor. An *agent* $\pi : \mathcal{S} \mapsto \Delta_{\mathcal{A}}$ is a stationary stochastic policy mapping states to a distribution over actions.[3] Its value at state $s$, $V^\pi(s) = \mathbb{E}_{P,\pi} \left[ \sum_{t=0}^{H-1} \gamma^t r(s_t, a_t) \,\middle|\, s_0 = s \right]$, is its expected discounted return starting at $s$. The objective in RL is to compute an *optimal policy*, which maximizes value over some initial state distribution $\nu_0$, i.e., $\pi^* \in \arg\max_\pi \mathbb{E}_{s_0 \sim \nu_0}[V^\pi(s_0)]$.

*Policy gradient (PG)* methods are a class of RL algorithms that directly optimize the policy to maximize expected return [Schulman et al., 2017, Sutton et al., 1999]. In LLM tasks, PG often uses a reference policy $\pi_{\text{ref}}$ as an anchor to regularize the RL task (e.g., stay close to $\pi_{\text{ref}}$). This is the prevalent approach in *reinforcement learning from human feedback* (RLHF, Ouyang et al. [2022]). In

---

[2]Further experiments on the Amazon review dataset [Ni et al., 2019] are provided in Appendix I.

[3]We limit ourselves to stationary policies for simplicity.

Table 1: Glossary

| | | | |
|---|---|---|---|
| $\mathcal{X}$ | Ambient space (e.g., descriptions of movies) | $d : \mathcal{X} \times \mathcal{X} \mapsto \mathbb{R}_+$ | Ambient metric |
| $\mathcal{Z}$ | Latent embedding space | $d_{\mathcal{Z}} : \mathcal{Z} \times \mathcal{Z} \mapsto \mathbb{R}_+$ | Latent metric |
| $\mathcal{A}$ | Action space (e.g., changes to a movie plot) | $U : \mathcal{Z} \times \mathcal{D} \mapsto \mathbb{R}_+$ | Utility function |
| $n$ | Latent dimension | $E_D : \mathcal{X} \mapsto \mathcal{Z}$ | Latent encoder |
| $\mathcal{D}$ | Dataset of entities | $\pi_{\text{ref}} : \mathcal{X} \mapsto \Delta_{\mathcal{A}}$ | Reference policy |
| $H$ | Horizon | $q : \mathcal{X} \mapsto \Delta_{\mathcal{A}}$ | reference policy distribution |

this work we achieve this using a KL-divergence loss over policy outputs; i.e.,

$$\text{loss}(\pi) = \text{PG-loss}(\pi) - \alpha D_{KL}(\pi || \pi_{\text{ref}}). \tag{1}$$

**Related Work.** Generative models are powerful tools for creative generation of text, images, and more [Achiam et al., 2023, Ho et al., 2020, 2022, Team et al., 2023]. Embedding representations [Arvanitidis et al., 2018, 2021, Oord et al., 2018, Yang et al., 2017] have been applied in many fields, including recommender systems [Liang et al., 2018], healthcare [Rampášek et al., 2019], neuroscience [Keshtkaran and Pandarinath, 2019], genetics [Frazer et al., 2021], astronomy [Ravanbakhsh et al., 2017], and more [Gómez-Bombarelli et al., 2018, Kingma et al., 2014, Thadani et al., 2023].

Aligning LLM generation with embedding spaces holds great potential. Our work constrains the generation process to work with a predefined utility over an embedding space. Other methods have been proposed to align language models to knowledge graphs [Bao et al., 2016], safety constraints [Ji et al., 2024, Yang et al., 2021], and human preferences [Bakker et al., 2022, Ouyang et al., 2022, Wang et al., 2024]. Our approach can be generalized to such tasks with a suitable latent embedding space and utility specification.

The use of RL with human and AI feedback for LLM fine-tuning has been shown to induce significant improvement in LLM capabilities [Lee et al., 2023, Ouyang et al., 2022]. Embedding-driven guidance is an additional form of feedback that can further improve creative LLM generation, and potentially reduce hallucination [Dhuliawala et al., 2023, Gunjal et al., 2024, Li et al., 2023].

## 3 Problem Setup

We assume an *ambient entity space* $\mathcal{X}$, which represents the space of all possible entities (e.g., all movie plots, or all animal photos). Since our setup is agnostic to the modality of the space $\mathcal{X}$, we do not emphasize language entities in this section. We assume $\mathcal{X}$ is equipped with a metric $d : \mathcal{X} \times \mathcal{X} \mapsto \mathbb{R}_+$, which may not be known. We also assume access to a *latent embedding* $E_D : \mathcal{X} \mapsto \mathcal{Z}$, which maps entities in $\mathcal{X}$ to a latent embedding space $\mathcal{Z} \subseteq \mathbb{R}^n$. If $E_D$ is injective, it induces a metric $d_{\mathcal{Z}}(z, w) = d(E_D^{-1}(z), E_D^{-1}(w))$ over $\mathcal{Z}$. Otherwise, we define the metric $d_{\mathcal{Z}}(z, w) = \sup_{x \in E_D^{-1}(z), y \in E_D^{-1}(w)} d(x, y)$ for $z \neq w$, and $d_{\mathcal{Z}}(z, z) = 0$.[4] Let $(\mathcal{Z}, d_{\mathcal{Z}})$ be the embedding manifold. Though $d_{\mathcal{Z}}$ is not known, various methods exist to estimate the underlying metric (e.g., see Appendix B on the use of Riemannian manifolds and pullback metrics of generative models to estimate $d$ [Arvanitidis et al., 2018, 2021]).

Given a fixed set of entities $\mathcal{D} = \{x_i \in \mathcal{X}\}_{i=1}^N$ of size $N$ (i.e., a dataset), we define a utility function $U : \mathcal{Z} \times \mathcal{D} \mapsto \mathbb{R}$ over embeddings on the manifold $(\mathcal{Z}, d_{\mathcal{Z}})$. Below we illustrate an example of a utility function for quantifying the quality of a *content gap*.

**Example: Content Gaps.** Consider a utility function $U$ measuring if a hypothetical point $z \in \mathcal{Z}$ is a content gap in a recommender ecosystem [Boutilier et al., 2024]. We might define $U$ as:

$$U(z; \mathcal{D}) = \underbrace{\sum_{\text{user}} U_{\text{user}}(z; \mathcal{D})}_{\text{Users' Utility}} + \underbrace{\sum_{\text{creator}} U_{\text{creator}}(z; \mathcal{D})}_{\text{Creators' Utility}} + \underbrace{\sum_{x \in \mathcal{D}} U_d(d_{\mathcal{Z}}(E_D(x), z))}_{\text{Distance from existing content}}. \tag{2}$$

Here $\mathcal{D}$ is the set of *existing* content items, and $U_{\text{user}}, U_{\text{creator}}, U_d$ are the utility functions for users, creators, and distance, defined over the latent embedding space. That is, a content gap $z$ should bring high utility to users and creators, while also meeting some latent demand (i.e., non-existing content). In Sec. 5 we explicitly formulate this utility for a behavioral latent embedding space trained over movie and user data.

---

[4] Other metric definitions are possible; we use this definition for illustration only.

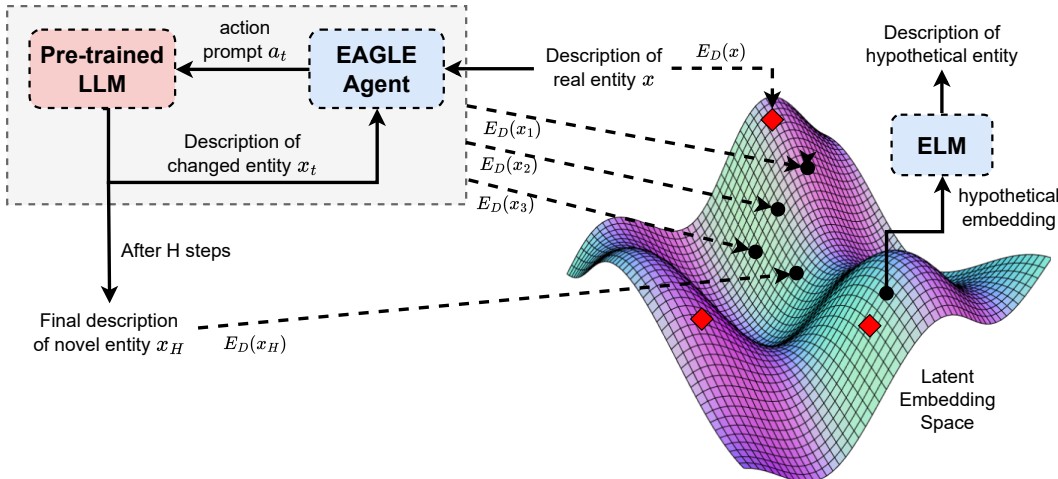

Figure 1: An illustration comparing the creation of descriptions of novel entities using ELM [Tennenholtz et al., 2024] vs. EAGLE (ours). The latent embedding space $\mathcal{Z}$ is illustrated as a complex surface on the right. Red points on the surface are illustrated as latent embeddings of existing entities. Black points are used to illustrate hypothetical (i.e., non-existing) entities. In ELM, a utility is maximized *in latent embedding space* to identify an optimal point. This hypothetical embedding is then decoded back to ambient space $\mathcal{X}$ to form a description of the hypothetical entity. Conversely, the EAGLE agent utilizes a highly capable pre-trained LLM as an environment to search for novel entities in ambient space $\mathcal{X}$. EAGLE does not use a decoder, but rather only requires an encoder $E_D : \mathcal{X} \mapsto \mathcal{Z}$. More specifically, the EAGLE agent uses action prompts to change an existing entity using the environment LLM. Every new changed entity is then mapped back to the latent embedding space using the encoder $E_D$. The utility function is optimized by the EAGLE agent through a reward signal to stir the changes to areas of high utility in the latent embedding space. A final description is then returned after $H$ steps.

**Optimality Criterion.** Given the dataset of entities $\mathcal{D} = \{x_i \in \mathcal{X}\}_{i=1}^N$, embedding manifold $(\mathcal{Z}, d_{\mathcal{Z}})$, a corresponding embedding function $E_D : \mathcal{X} \mapsto \mathcal{Z}$, and a utility function $U : \mathcal{Z} \times \mathcal{D} \mapsto \mathbb{R}$, our goal is to find a novel entity $x^* \in \mathcal{X} \backslash \mathcal{D}$ such that

$$x^* \in \arg\max_{x \in \mathcal{X}, x \notin \mathcal{D}} U(E_D(x); \mathcal{D}). \tag{3}$$

We emphasize that, while optimality is defined in the *latent embedding* space, our goal is to identify a novel entity $x^*$ in the *ambient* space $\mathcal{X}$ (e.g., an image or description of the hypothetical). As such, it is not enough to search for an optimal point in $\mathcal{Z}$. Moreover, the solution is not necessarily unique.

## 4   Surfacing Optimal Entities Using LLMs

We now turn to generation of novel entities in language space; that is, we assume $\mathcal{X} \subseteq \mathcal{S}$ is a subset of the space of all language sequences (e.g., plots of movies, or descriptions of clothes). Nevertheless, our methods are applicable more broadly to other modalities including images, videos, audio, or even RL policies. In what follows we describe two methods of finding an optimal entity $x^*$ in Eq. (3), using either ELMs or RL. A high-level illustration of these methods is shown in Fig. 1.

### 4.1   Embedding Language Models (ELMs) as Decoders

ELMs (Sec. 2) offer an explicit method of finding an optimal entity $x^* \in \mathcal{X}$. Indeed, if one can train a decoder $f : \mathcal{Z} \mapsto \mathcal{X}$ which satisfies $E_D(f(z)) = z$, then $x^*$ can be determined by maximizing $z^* \in \arg\max_{z \in \mathcal{Z}} U(z; \mathcal{D})$ (i.e., identifying an optimal latent embedding w.r.t. $U$) and returning $x^* = f(z^*)$ (i.e., some entity corresponding to $z^*$). Such a decoder $f$ can be trained using a fixed dataset and an ELM architecture [Tennenholtz et al., 2024].

However, this approach has major limitations. First, learning an ELM decoder $f$ requires training an LLM to interpret embeddings, a costly process which may negatively impact the expressive power of the LLM. Second, the manifold $(\mathcal{Z}, d_{\mathcal{Z}})$ can be highly complex and non-linear. Moreover, the latent space metric $d_{\mathcal{Z}}$ may not be known. As such, a latent embedding $z^*$ deemed optimal in latent

space, may be "off-manifold", and therefore not mapped to a valid entity $x \in \mathcal{X}$. To mitigate this, one can choose to stay close to points in the data, using Euclidean distance to approximate $d$ [Chen et al., 2019], or employing a pullback metric defined by a Riemannian manifold [Arvanitidis et al., 2018, 2021] (see Appendix B for further discussion). Finally, the embedding space $\mathcal{Z}$ may not contain enough semantic information to ensure that $f$ generalizes well (e.g., embeddings trained over behavioral interactions with items). Training a new embedding, e.g., using Variational Auto-Encoders (VAEs) [Yang et al., 2017] can mitigate this problem; yet, this would require training a new latent embedding, a challenging problem in its own right [Vahdat and Kautz, 2020]. Furthermore, training a new latent representation may not be feasible (e.g., due to limited data). As such, using ELMs to train a decoder LLM over the embedding space may be impractical in many domains.

## 4.2 A Reinforcement Learning Perspective

As an alternative to ELMs, we formulate the optimization directly in the ambient space $\mathcal{X}$ (i.e., language space) using RL. Specifically, we leverage a highly capable pre-trained LLM (e.g., Gemini-Ultra [Team et al., 2023], or GPT-4 [Achiam et al., 2023]) to act as an *environment* for our RL formulation. The environment LLM remains fixed, and is only used to generate new entities in $\mathcal{X}$.

We formulate the problem using an MDP $(\mathcal{X}, \mathcal{A}, P, r, H)$, where $\mathcal{X} \subseteq \mathcal{S}$ is the state space comprising of all possible entities (in language), $\mathcal{A} \subset \mathcal{S}$ is an action space, defined by language prompts for a pre-trained environment LLM.[5] The action prompts are used to modify or *steer* an entity $x \in \mathcal{X}$. We elaborate on these actions later in this section. The transition function, $P : \mathcal{X} \times \mathcal{A} \mapsto \Delta_{\mathcal{X}}$ is induced by the environment LLM. Finally, the reward function is defined using the utility $U$ such that $r_t(x) = 0$ for all $t < H - 1$, and $r_{H-1}(x) = U(E_D(x); \mathcal{D})$.[6] While we do not define it explicitly, the MDP can also be endowed with contextual information such as information about nearby entities, or a user for whom we wish to personalize creation.

At time $t = 0$ the process is initialized at an existing entity $x_i \in \mathcal{D}$. For every $t < H$, the agent sees state $x_t \in \mathcal{X}$ and selects an action $a_t \in \mathcal{A}$. The action is provided to the environment LLM as a language prompt, the environment LLM generates the next state $x_{t+1} \in \mathcal{X}$, and the agent receives a reward $r(x_t)$. We assume the environment LLM generates states in $\mathcal{X}$ (i.e., feasible states).

## 4.3 Designing a State-Dependent Action Space

Designing an action space for an RL problem can be challenging [Guss et al., 2019, Kanervisto et al., 2020, Tennenholtz and Mannor, 2019]. In our setting, actions are prompts used to change entities in $\mathcal{X}$. As such, a good set of actions should be diverse and induce high utility states (entities).

**LLMs for Action Creation.** We exploit the generative power of LLMs to construct an exploratory set of actions for each state. Specifically, for each $x \in \mathcal{X}$, an LLM is used to construct a set $\mathcal{A}(x)$ of $K$ actions. For example, we might prompt an LLM to generate 100 ways to change a plot of a movie, or the specification of an designed product. This ensures each existing $x \in \mathcal{D}$ is associated with a diverse set of actions that can modify it. To ensure coverage, $K$ must be large enough to adequately search the state space (see Sec. 7 and appendix C further discussion).

**Reference Policy and G-Optimal Design.** We use the (possibly large) set of actions to learn a reference policy for a PG algorithm (see Sec. 2), trained via supervised learning. Specifically, given the action space, we define a *reference* policy $\pi_{\text{ref}}(a|x) = q_a$ where $q \in \Delta(\mathcal{A}(x))$.

We consider three choices for distribution $q$: (1) uniform, i.e., $q_a = \frac{1}{K}$; (2) myopic best next-step action, i.e., $q_a = \mathbb{1}\{a \in \arg\max \mathbb{E}_{x' \sim P(\cdot|x,a)}[U(x')]\}$; and (3) myopic *G-optimal design*, which is a distribution over actions that minimizes a form of worse-case variance [Atwood, 1969, Kiefer and Wolfowitz, 1960]. We define it formally as follows.

**Definition 1** (G-optimal design). *Let $\mathcal{A}(x) \subseteq \mathcal{A}$ be given. Denote $z_a(x) = \mathbb{E}_{x' \sim P(\cdot|x,a)}[E_D(x')]$. A distribution $q(x) \in \Delta(\mathcal{A}(x))$ is a myopic G-optimal design with approximation factor $C \geq 1$ if*

$$\sup_{a \in \mathcal{A}(x)} \|z_a(x)\|^2_{\Sigma(q(x))^{-1}} \leq Cn, \tag{4}$$

*where $\Sigma(q(x)) = \mathbb{E}_{a \sim q(x)}[z_a(x)z_a^T(x)]$, and $n$ is the dimension of the latent embedding space $\mathcal{Z}$.*

---

[5]We use the term pre-trained LLM to refer to an already trained LLM such as a capable Gemini or GPT model. We do not refer here to the specific stage in LLM training known as "pre-training".

[6]More generally, intermediate rewards can be provided at every iteration. We do not use such rewards here as we focus only on the final generated outcome.

---

**Algorithm 1** Embedding-Aligned Guided LanguagE (EAGLE) Agent

---
1: **Input:** Environment LLM, dataset $\mathcal{D}$, reward function $r$, encoder $E_D$, PG algorithm `PG-ALG`
2: Generate $K$ candidate actions using pre-trained environment LLM for every $x \in \mathcal{D}$.
3: Compute an approximate myopic G-optimal design $q(x)$ for every $\mathcal{A}(x), x \in \mathcal{D}$ (Definition 1).
4: Train reference policy $\pi_{\text{ref}}(\cdot|x) = q(x)$ over candidate action data and G-optimal design.
5: Train `PG-ALG` with reference policy $\pi_{\text{ref}}$, pre-trained environment LLM, encoder $E_D$, and reward $r$.

---

G-optimal designs have proven useful for contextual bandits with large action spaces, allowing one to achieve efficient regret guarantees with minimal assumptions [Zhu et al., 2022]. Notably, an optimal design as a uniform $\epsilon$-exploration strategy has been shown to produce an efficient algorithm. We implement such exploration using a G-optimal design-based reference policy. A G-optimal design always exists whenever $\{z_a(x)\}_{a \in \mathcal{A}(x)}$ is compact, and $C = 1$ [Kiefer and Wolfowitz, 1960, Zhu et al., 2022]. While it is generally NP-hard to compute [Grötschel et al., 2012], an approximate optimal design can be computed efficiently [Zhu et al., 2022] (see Appendix C for specific implementation details in our setup using uniform sampling).

### 4.4   Embedding-Aligned Guided LanguagE (EAGLE) Agent

We are now ready to describe our approach, the Embedding-Aligned Guided LanguagE (EAGLE) agent, which is detailed in Algorithm 1. An illustration comparing EAGLE to ELM is depicted in Fig. 1. The EAGLE agent is trained using the pre-trained environment LLM. We first generate $K$ actions using the dataset of existing entities $\mathcal{D}$ and the pre-trained LLM. Then, using the encoder $E_D : \mathcal{X} \mapsto \mathcal{Z}$, we create an approximate myopic G-optimal design $q$ over these actions. We use the G-optimal design to train an initial reference policy $\pi_{\text{ref}}$ to match the action distribution $q$. Finally, we use $\pi_{\text{ref}}$ as a reference policy for a PG algorithm, regularizing the loss as in Eq. (1). The PG algorithm is trained using the environment LLM and the encoder $E_D$ to optimize a utility $U$.

## 5   Experiment Design

We detail our experiment design; specifically, the data generation process, the training process, and our evaluation methods. Our work uses the MovieLens 25M dataset [Harper and Konstan, 2015], which contains 25 million ratings (1 to 5) of 62,423 movies and 162,541 users. We also conduct experiments on the Amazon review dataset [Ni et al., 2019], whose results are provided in Appendix I.

### 5.1   Data Generation

**Latent Embeddings.** We create embeddings for both users and movies in the MovieLens dataset. We consider *behavioral embeddings*, trained solely using user ratings (i.e., no explicit semantic information) via *matrix factorization* using *weighted alternating least squares* [Hu et al., 2008]. A user $u$'s predicted rating for movie $m$ is given by the dot product $\langle z_u, z_m \rangle$ of their embeddings.[7]

**State Space.** We generate textual descriptions of movies and users in the MovieLens dataset using Gemini Ultra [Team et al., 2023]. For movies, we prompt Gemini to generate a plot for the movie, list five reasons someone might like the movie, and list five reasons they might dislike it (we refer to these as the *description* of a movie). To describe a user $u$, we provide Gemini with the list of movies $u$ rated with their ratings, and prompt Gemini to create a textual *user profile* describing the user's preferences using a structured template with the following categories: general, genre, director, and aesthetic preferences. Examples of generated movie descriptions, user profiles, and prompts are provided in Appendices D, F and H. To test the consistency of the generated profile for a user $u$, we train an encoder to map it to $u$'s behavioral embedding (see Appendix D).

**Action Space.** We generate 100 candidate actions for each movie in the dataset. Specifically, we generate 50 generic actions and 50 personalized actions for every movie. We do this by prompting Gemini Ultra with the plot of the movie, and the list of reasons someone might like or dislike it,

---

[7]Other collaborative filtering methods could be used, e.g., dual encoders [Yang et al., 2020, Yi et al., 2019], but our approach is agnostic to the precise method used to generate behavioral embeddings.

and ask it to generate 50 actionable ways in which to change the movie. Similarly, to generate personalized actions, we also provide Gemini with a textual user profile, and ask it to generate 50 changes that are personalized to that user. To increase action set diversity, we prompt the model to output actions in five categories: plot changes, character enhancements, visual and storytelling improvements, thematic exploration, and audience appeal adjustments. See Appendices C, F and H for the precise prompts and sample outputs.

Notice that we rely on an LLM's ability (in our case, Gemini Ultra) to generate diverse actions that suffice to explore the ambient space of movies. Other domains may require further considerations to ensure action space diversity (see further discussion of this limitation in Appendix B).

## 5.2 Training

**Training Task.** We consider a specialized content gap task described in Sec. 3. We simplify somewhat by not accounting for creator utility, and only considering "local" content gaps around anchor items and specified users. We compute user utility for items in behavioral embedding space using dot products of user and movie embeddings, $\langle z_u, z_m \rangle$. We ensure the generated movie is, in fact, a content gap by also accounting for distance to existing movies; specifically, $\ell_2$ distance (in embedding space) of the generated movie to its three nearest neighbors. Formally, given user $u$ with embedding $z_u$, and a target movie $m \in \mathcal{D}$ with embedding $z_m = E_D(m)$, our utility is defined by

$$U(z_m; \mathcal{D}) = \langle z_u, z_m \rangle + \lambda \sum_{m' \in \text{NN}(m)} \|z_m - z_{m'}\|, \qquad \text{(Content Gap Utility)}$$

where $\text{NN}(m)$ is the set of three nearest neighbors to $m$ in $\mathcal{D}$ (w.r.t. $\ell_2$ distance in $\mathcal{Z}$), and $\lambda > 0$.

**Encoder.** We fine-tune a Gemini Nano-2 LLM (3.25B parameters) [Team et al., 2023] to serve as a movie encoder $E_D$. We train it using $\ell_2$ regression loss to map movie descriptions (plot, reasons to like, and reasons to dislike) to their behavioral embeddings. We use $E_D$ to compute the utility $U$.

**Reference Policy.** We use Gemini Nano-2 to initialize a reference policy $\pi_{\text{ref}}$. We fine-tune the model using next-token prediction (cross-entropy loss) of actions as targets. We create three reference policies: *(a) Uniform:* We use the full set of 100 generated actions for each movie as targets. *(b) Best next-step action:* We first create transitions for each action in the dataset. Specifically, we use Gemini Ultra to act as an environment, and prompt it with each action to generate a new movie (plot and reasons to like or dislike). The new movie is then encoded using $E_D$, and its utility computed. Finally, we create a dataset of the "best" actions based on the computed scores. *(c) Approximate G-optimal design:* We calculate the next step embeddings for each action, as before. We then select ten actions which are a solution to Eq. (4). The new dataset consists of ten "exploratory" actions per movie, which we use to train a reference policy via supervised learning, as before. We refer to Appendix C for details on computing the optimal design.

**RL Training.** We use Gemini Ultra as our environment LLM. It is prompted to generate a transition given the current movie description and the agent action (see Appendix F for a review of all prompts). We inject randomness by increasing the sampling temperature of the agent model and the environment LLM. Specific hyperparameters are given in Appendix E.

## 5.3 Evaluation Methods

We employ two evaluation methods; namely, raw utility scores and human feedback, comparing EAGLE-generated outputs with their initial anchors. We use a pool of 124 human raters to provide feedback on our generated results.[8] Raters are shown a textual user profile together with descriptions of two movies: a ground truth (real) movie, and a variant generated using $H = 5$ steps of agent changes. Importantly, the two descriptions are ordered randomly, and raters do not know which is the original or generated variant.

We quantify each rater's personal preferences by having them rate 10–20 movies from a set of 600 popular movies, and ask them to write a textual profile of their own preferences. To increase confidence in our human evaluation, raters evaluate each algorithm twice, once w.r.t. a textual user

---

[8]Raters were paid contractors. They received their standard contracted wage, which is above the living wage in their country of employment.

Table 2: Comparison of ELM, supervised training, and EAGLE. We test three distribution of reference policies: uniform (i.e., random), optimistic (i.e., next-step best action), and G-optimal design (Definition 1).

| Experiment | | Utility | Human Rater Evaluation | | |
|---|---|---|---|---|---|
| | | $U(z)$ | User Utility | Rater Utility | Distance Score |
| ELM [2024] | | 0.57 | 0.31 | 0.37 | **0.52** |
| Reference Policy | Uniform | $0.49 \pm 0.02$ | 0.59 | 0.62 | 0.44 |
| | Optimistic Action | $0.58 \pm 0.02$ | 0.62 | 0.6 | 0.34 |
| | G-Optimal Design | $0.51 \pm 0.02$ | 0.51 | 0.67 | 0.45 |
| EAGLE (ours) | Uniform | $0.65 \pm 0.03$ | 0.69 | 0.64 | 0.43 |
| | Optimistic Action | $0.69 \pm 0.04$ | 0.71 | 0.69 | 0.25 |
| | G-Optimal Design | $0.74 \pm 0.03$ | **0.76** | **0.74** | 0.47 |

Table 3: For `EnvTrain` $\in$ {Gemini Pro, Gemini Ultra} and `EnvTest` $\in$ {Gemini Ultra, GPT-4}, the experiment `EnvTrain`→`EnvTest` shows results for training EAGLE on `EnvTrain` and testing it on `EnvTest`. All experiments use an EAGLE based on Gemini-Nano and a G-optimal design baseline.

| Experiment | Utility | Human Rater Evaluation | | |
|---|---|---|---|---|
| | $U(z)$ | User Utility | Rater Utility | Distance Score |
| Gemini Pro → Ultra | $0.75 \pm 0.03$ | 0.76 | 0.74 | **0.47** |
| Gemini Ultra → Ultra | $\mathbf{0.8 \pm 0.02}$ | 0.74 | 0.66 | **0.47** |
| Gemini Pro → GPT-4 | $0.65 \pm 0.02$ | **0.86** | **0.78** | **0.45** |

profile of a user *from the MovieLens dataset*, and again w.r.t. *their own* preference profile. Each rater is asked a series of questions to quantify: (1) *User Utility*, the quality of the generated movie w.r.t. a user profile; (2) *Rater Utility*, its quality w.r.t. their personal preferences; and (3) *Distance Score*, the distance of the generated movie from the anchor (i.e., the rater scores how different the two movies are). We express our results as the percentage of raters who preferred the generated result to the original (values in $[0, 1]$). See Appendix G for a complete overview of rater questions.

# 6 Experiments

Table 2 compares results of ELM, supervised training (i.e., the reference policy), and EAGLE (ours) using the three reference action distributions (see Secs. 4.3 and 5.2). Specific hyperparameters are detailed in Appendix E. We train ELM to map behavioral embeddings of existing movies to their descriptions. Given an input movie embedding, we search for a nearby embedding point with optimal utility, and use ELM as a decoder to produce its description. We see that EAGLE outperforms ELM by a large margin w.r.t. both user and rater utility. This suggests a limitation in the generalization capability of ELM, and a fundamental problem of moving "out of manifold" when searching for an optimal point in embedding space. We discuss this further in Appendix B. Despite this, raters judge ELM to generate movie descriptions that are further from (more different than) the anchor.

The reference policy performs best with optimistic actions (i.e., max next-step utility). Interestingly, even the randomized policies—uniform and G-optimal design—perform well. We believe this is due to the choice of action space, which includes many personalized actions, providing a strong initialization for the randomized policies. Moreover, EAGLE performs better with a G-optimal design baseline than with the uniform or optimistic action baselines. This suggests that G-optimal design helps search the state space more efficiently. This is also evident in the distance score, which increases with the use of G-optimal design.

**Environment Transfer.** Table 3 compares results for different training/inference setups to test transfer effectiveness. Specifically, we compare training with Gemini-Pro or Gemini-Ultra models, and testing (i.e., inference) with Gemini-Ultra or GPT-4 models. All results use EAGLE with a G-optimal design baseline. Our results suggest that a Gemini-Pro training environment suffices to obtain high-quality inference results, as training with Gemini-Ultra did not improve performance.

Table 4: The effect of changing the action space on EAGLE, showing (1) the default action space as described in Sec. 5, (2) the default action space with personalized actions removed, (3) only macro actions (i.e., taking three actions at once), and (4) a combined action space, with both the default actions as well as macro actions. All experiments use a G-optimal design strategy as reference policy.

| Experiment | Utility | Human Rater Evaluation | | |
|---|---|---|---|---|
| | $U(z)$ | User Utility | Rater Utility | Distance Score |
| Default Actions | **0.75 ± 0.03** | 0.76 | 0.74 | **0.47** |
| Without Personalized | 0.56 ± 0.02 | 0.54 | 0.59 | **0.45** |
| Macro Actions | 0.71 ± 0.02 | 0.67 | 0.7 | 0.32 |
| Combined Actions | **0.74 ± 0.04** | **0.88** | **0.92** | 0.4 |

Interestingly, a trained EAGLE agent tested on a fixed GPT-4 environment maintains (and sometimes increases) its level of performance, suggesting that EAGLE is robust to the LLM environment on which it acts. More broadly, it seems that one can train an EAGLE agent using any suitably capable LLM, and then apply it successfully in environments defined by different LLMs.

**Action Space.** We test the effect of changing the resolution and personalization of the action space on an EAGLE agent with a G-optimal design baseline in Table 4. We first test the effect of making personalized actions unavailable. As expected, we see significant degradation in performance, suggesting personalized actions are critical to performance.

Second, we construct a *macro-action space* comprising tuples $(a_1, a_2, a_3, a_4, a_5)$ of five consecutive actions. Thus, instead of making a single change to a movie at each step, the environment LLM makes five changes at once (i.e., $H = 1$). We evaluate the macro action space by executing the five actions chosen by the trained EAGLE agent on the default action space, at once. We also construct a *combined action space*, that *adds* macro-actions to the original action space (rather than replacing it). We see that macro-actions degrade performance w.r.t. the original finer-grained changes, suggesting the LLM performs better when making smaller changes. We also find the combined action space significantly outperforms other baselines, suggesting value to equipping the agent with both low and high resolution actions in a sequential setup.

**EAGLE on Highly Rated Movies.** Finally, we assess the conditional performance of the EAGLE agent over movies that are originally rated high. We find that, while EAGLE improves overall scores of anchor movies that are originally rated poorly (i.e., scores $1, 2, 3$), it does not perform as well on movies that are rated highly (e.g., scores $4, 5$), sometimes even decreasing user utility. Specifically, conditioned on poorly rated movies, EAGLE achieves an average utility of $0.7 \pm 0.04$ (an approximate $30\%$ increase), whereas for highly rated movies, it achieves an average utility of $0.83 \pm 0.03$ (an approximate $5\%$ decrease). This result is expected as MovieLens rating data is limited to ratings between $1$ and $5$, where a rating of $5$ is (perhaps, incorrectly) assumed to be optimal for the user. To address this, we might only suggest or create a new entity if it has a higher utility than its anchor.

# 7 Creative Generation: Discussion and Limitations

**ELM vs. EAGLE.** Our work reveals two distinct methodologies for generating novel entities that adhere to an underlying embedding space: using ELMs as decoders and our proposed EAGLE agent. While ELM offers efficient optimization within the embedding space and, in theory, can reach an unconstrained optimal point, it faces challenges related to the computational cost of fine-tuning the decoder, unknown generalization errors, and difficulties in constraining optimization due to the unknown manifold metric. Conversely, EAGLE leverages the existing textual capabilities of LLMs, enabling more interpretable optimization and potentially requiring a smaller agent model. However, EAGLE's coverage is inherently limited by the defined action space, and training can be computationally expensive due to its reliance on LLM environment queries. We refer the reader to Appendix B for further discussion on these tradeoffs.

**Action Space Design.** A principal building block of EAGLE is defining and / or designing an action space to change entities in $\mathcal{X}$. Designing a good action space is critical to ensure reachability

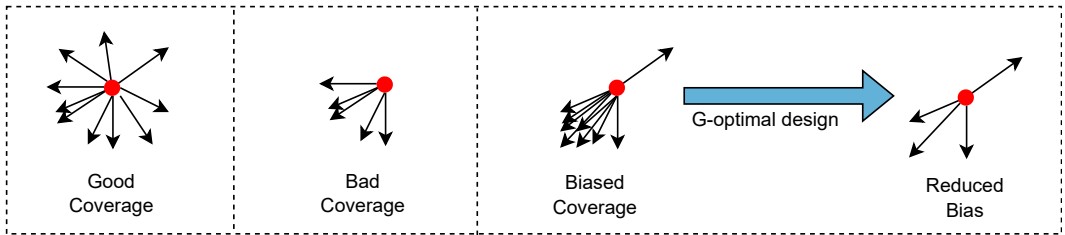

Figure 2: An illustration of different forms of coverage and bias of an action space. As actions are textual prompts, their corresponding embedding directions may either provide good coverage, or partial coverage of the underlying embedding manifold (e.g., there may be directions that are not covered by the generated actions). Moreover, actions may be uniformly biased toward specific directions. We accommodate for this using a G-optimal design which reduces this bias through a set of exploratory actions in embedding space.

of an optimal novel entity. In this work we define a methodology for designing such an action space using LLMs and G-optimal design (Definition 1).

Fig. 2 illustrates three examples of potential action spaces and their inherent bias in embedding space. Informally, an action space may allow for "better" coverage if it can induce changes in all direction on the embedding manifold $\mathcal{Z}$. More generally, an action space should allow one to reach any point on the embedding manifold (up to some ball of radius $\epsilon$) in at most $H$ iterations, starting from any point $z = E_D(x)$ such that $x \in \mathcal{D}$. Indeed, as we illustrate in Fig. 2, our action space may be constrained to certain directions, thereby limiting our ability to cover the full span of possible entities.

To increase the overall coverage of an action space one can take one of several measures, including: (1) generating a large amount of candidate actions, (2) using experts to enhance the action space with additional missing actions, or (3) changing the methods used to generate those actions (e.g., using different prompting strategies). While all of these methods could potentially increase the coverage of the action space, none of them would guarantee coverage of the embedding space. In fact, it may be that *no action exists* that would allow one to move in any direction of the embedding space. That is, the environment LLM is incapable of moving in a specific embedding direction (i.e., an embedding direction cannot be mapped into an actionable text prompt). This is a particular disadvantage of EAGLE, and an advantage of ELM, as ELM is not constrained by movements in embedding space (see Appendix B).

Apart from the coverage challenge of designing an action space, we also consider the bias of such an action space. If an action space is biased, then a randomized policy executing those actions uniformly would be biased as well. To mitigate this bias effect we use a G-optimal design. The G-optimal design allows us to identify a set of $k$ actions that would be best for exploring the action space in terms of overall utility, thereby mitigating the inherent bias for exploration.

# 8 Conclusion

This paper introduces EAGLE, a novel framework that leverages RL to align LLM generation with domain-specific objectives encoded in latent embedding spaces. Treating a pre-trained LLM as an environment, EAGLE iteratively steers its output towards optimal regions of the embedding space based on a predefined criterion. Our experiments on the MovieLens 25M dataset demonstrate the effectiveness of EAGLE in surfacing content gaps, hypothetical content items that satisfy latent user demand. The results highlight the benefit of incorporating a state-dependent action set based on G-optimal design for enhanced efficiency. EAGLE's capacity to leverage the power of large pre-trained LLMs without requiring explicit decoders opens up new possibilities for controlled and grounded text generation, ensuring consistency with domain knowledge and data representations.

Further research directions include: exploring different action space design methodologies beyond G-optimal design, generalizing EAGLE to diverse modalities like images, audio, and video, and applying EAGLE to other applications that require aligning LLM generation with specific domain knowledge or constraints. EAGLE's flexible framework holds promise for various tasks, including personalized content creation, targeted advertising, and controlled dialogue generation.

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

# A  Societal Impact

EAGLE, a tool capable of creating new content based on latent preferences, presents several societal implications. On one hand, EAGLE can enhance creativity and innovation by assisting content creators in exploring new ideas and generating novel content tailored to specific audiences. This can stimulate creative industries and lead to the development of more diverse and engaging content. Additionally, by identifying and surfacing content gaps, EAGLE can help address unmet needs and preferences within online platforms, leading to a more inclusive and satisfying user experience. Furthermore, EAGLE facilitates the generation of personalized content tailored to individual users, potentially improving engagement and satisfaction.

However, the technology also carries potential risks. If trained on biased data, EAGLE could amplify existing societal biases and reinforce echo chambers, limiting exposure to diverse perspectives. The ability to tailor content to latent preferences could also be misused to manipulate user behavior, leading to unintended consequences.

Mitigating these risks requires a multi-faceted approach. Ensuring data diversity and fairness during training is crucial to mitigate bias and promote inclusivity. Developing methods to make EAGLE's decision-making process transparent can help address concerns about manipulation and promote user trust. Finally, integrating fact-checking mechanisms and robust content moderation strategies can help prevent the spread of misinformation.

Developing and deploying EAGLE demands careful consideration of its potential societal impacts. Proactive measures to mitigate risks and promote responsible use are crucial to harnessing its potential benefits while minimizing potential harm.

# B  ELM or EAGLE: Trade-Offs and Limitations

As discussed in Sec. 4, ELM and EAGLE can be viewed as alternative approaches for solving similar problems involving embedding spaces. Particularly, when we care about generating entities which adhere to some underlying embedding space. While ELM assumes the existence of a decoder $f : \mathcal{Z} \mapsto \mathcal{X}$, EAGLE only requires access to an encoder $E_D$ to assess the quality of any $x \in \mathcal{X}$. Nevertheless, as we ultimately care about generating non-existing entities $x \notin \mathcal{D}$, we must ask ourselves – how do we find an optimal $x$ in a generalizable and robust manner. Below, we enumerate several of the trade-offs of using ELMs and EAGLE to solve this problem, focusing on three points: computational efficiency, realizability, and coverage. We summarize these trade-offs in Appendix B.

**Computational Efficiency.** When considering computational efficiency of ELM compared to EAGLE we account for training efficiency, inference efficiency, as well as the computational requirement to retrain the model over time (due to e.g., evolving data and metrics).

We find that ELM generally requires fine-tuning a much larger model than the EAGLE agent. Particularly, ELM should be capable of generating a full description of any embedding point in $\mathcal{Z}$. This would potentially require training / fine-tuning a high capacity model. On the other hand, EAGLE requires generating action prompts, which are shorter in nature, and, as we find, can use much smaller models. Still, during training, EAGLE requires querying an environment LLM, which potentially uses many computational resources. Moreover, in our work we use 16 environment LLMs simultaneously to train the EAGLE agent, requiring extensive compute during training. That said, the environment LLM can in practice be an API call to a service which, unlike ELM, does not require serving an LLM in-house.

Finally, we note that ELM is more sensitive to changes in the dataset and metrics. As we see in Table 3, EAGLE is robust to changes in the environment LLM. More broadly, as EAGLE uses action prompts to change existing entities, much of the underlying "ground work" is done by the environment LLM. Therefore, if the dataset and metric were to change, e.g., one would want to also personalize outputs to specific creators, add more context to the generation, or perhaps generate other attributes of the hypothetical entities, then EAGLE would merely need to change the prompt of the environment LLM, whereas ELM would require retraining the model over new data to account for these changes.

The computational aspect of using ELM vs. EAGLE thus depends on the use case, the availability of resources to train LLMs (or using API calls), and the evolution of the underlying dataset and metrics.

Table 5: Trade-offs between EAGLE and ELM

| Method | Advantages | Disadvantages |
|--------|-----------|---------------|
| **ELM** | • Efficient optimization in embedding space.
• Not constrained - can theoretically reach optimal point in embedding space. | • Computational efficiency of fine-tuning an ELM.
• Unknown generalization error / manifold metric for constraining optimization. |
| **EAGLE** | • Leverages textual proficiency of existing LLMs.
• Interpretable optimization
• Computational efficiency: agent can be implemented using a smaller model. | • Coverage is constrained by action space.
• Computational efficiency during training: must use LLM environment. |

**Realizability.** A critical aspect when comparing ELM to EAGLE is the realizability of the underlying embedding manifold. Even if ELM manages to learn a perfect decoder $f : \mathcal{Z} \mapsto \mathcal{X}$, there remains the question of identifying an optimal point $z \in \mathcal{Z}$ to decode. As we only have embedding points of real entities $x \in \mathcal{D}$, extrapolating to non-existing entities requires moving on the embedding manifold $\mathcal{Z}$, which is generally unknown. If we were to simply search for a point $z \in \mathcal{R}^n$ which maximizes utility, we may return a point that does not correspond to a real entity, and thus decoding would not be feasible. Moreover, when as decoder is trained on existing entity data $\mathcal{D}$, one must also account for generalization errors when attempting to move out of distribution.

v We propose two solution to account for this problem when using ELMs. The first, is to constrain the search in embedding space to be close to points in the data, that is, balls of radius epsilon around all embedding points in $\mathcal{D}$:

$$\bigcup_{x \in \mathcal{D}} B_\epsilon(E_D(x)).$$

This solution allows one to confidently remain on manifold, with good generalization capabilities, while still optimizing for better content. Nevertheless, it may still be limited, as certain areas of the embedding space may be easier to decode than others.

Another approach to search over the manifold $(\mathcal{Z}, d_{\mathcal{Z}})$ is to use the geometry of a generative model [Arvanitidis et al., 2018, 2021] and estimate the metric $d_{\mathcal{Z}}$. Particularly, assuming the embedding space was trained by a stochastic generative model (such as a VAE), one can use the pull-back metric (defined by the decoder Jacobian) to define the geometry of the latent space. Specifically, for VAEs with a decoder of the form $f(z) = \mu(z) + \sigma(z) \cdot \epsilon$, such that $\mu : \mathcal{Z} \mapsto \mathcal{X}, \sigma : \mathcal{Z} \mapsto \mathbb{R}_+^{\mathcal{X}}, \epsilon \sim \mathcal{N}(0, I)$, this induces a Reimannian sub-manifold, with an expected metric

$$\mathbb{E}[M_z] = \left(J_z^{(\mu)}\right)^T \left(J_z^{(\mu)}\right) + \left(J_z^{(\sigma)}\right)^T \left(J_z^{(\sigma)}\right).$$

This has been shown to produce improved interpolations of embeddings, which better align with the underlying geodesics of the manifold [Arvanitidis et al., 2018, 2021]. Still, this approach is computationally expensive, as it requires computing the Jacobian of the decoder, which, in our case, consists of an LLM.

EAGLE achieves realizability of the embedding by construction. Assuming the environment LLM generate feasible descriptions in $\mathcal{X}$, any such point readily maps to a feasible embedding point. Moreover, training an encoder $E_D$ is often a significantly easier task than training a decoder. Indeed, we find our encoder generalizes well on the test set of movie descriptions. More importantly, using the encoder to map to the embedding manifold we ensure any generated point is realizable, overcoming this problem by "moving" in ambient space $\mathcal{X}$ instead of latent embedding space $\mathcal{Z}$. This, however, comes with a significant limitation, i.e., coverage, as we discuss next.

**Coverage.** One of the major limitations of working in ambient space $\mathcal{X}$, as opposed to directly optimizing the embedding space $\mathcal{Z}$ is the inherent coverage of actions in $\mathcal{X}$. Particularly, EAGLE requires using action prompts to change an entity's description. This approach has several disadvantages as (1) one must define the set of actions, (2) it is unclear the full embedding space can be

realized by textual action changes to existing entities, and (3) the action space may be highly biased to specific changes. Indeed, ELM is by design not constrained by this problem (though it does have the realizability challenge we discussed above).

We attempt to overcome some of these challenges in our work by using an LLM to generate a large pool of action candidates. Then, to remove potential bias and increase coverage we use a G-optimal design. Nevertheless, this process does not completely solve the problem. It may be that a combined ELM-EAGLE approach would be the most beneficial solution to the coverage-realizability tradeoff.

# C    Action Set Design

In our work all actions were generated by a Gemini Ultra. The following prompt was used to generate the set of personalized actions (a similar prompt was used for non-personalized actions):

---

### Prompt for Generating Personalized Actions

You are a producer of a movie. You will be given a description of a movie and a profile of someone who might want to watch such a movie.
Given the movie description and the user profile you will be asked to provide 50 actionable ways to change the movie.

Here is an example for the movie Inception (2010):

## BEGIN EXAMPLE

# Example Plot:
Inception follows Dom Cobb, a skilled thief who has the ability to enter people's dreams and steal their secrets. He is hired by businessman Saito to implant an idea into the mind of his competitor, Robert Fischer Jr. To accomplish this, Cobb assembles a team of specialists, including the architect Ariadne, the forger Eames, the chemist Yusuf, and the point man Arthur.
The team enters Fischer's dreams through a multi-layered process, each layer representing a deeper level of his subconscious. As they navigate the treacherous dream landscapes, they encounter resistance from Fischer's subconscious projections. Cobb also faces his own personal demons, including his deceased wife Mal, who manifests as a saboteur in the dreams.
In the deepest dream layer, Cobb and Ariadne confront Fischer and successfully implant the idea. However, Mal's projection attacks Cobb, forcing Ariadne to kill her. The team escapes the dream layers, but Saito is left trapped in limbo, the deepest dream state. Cobb enters limbo to retrieve Saito, promising him a return to reality.
Ultimately, Cobb and Saito return to the real world, and the inception is believed to be successful. However, the ending leaves an ambiguity as to whether Cobb himself is still dreaming.

# Why Some People May Like It:
- Innovative Concept: The idea of entering and manipulating dreams is a fresh and intriguing premise.
- Mind-Bending Visuals: The film's stunning special effects and imaginative dream sequences create a visually immersive experience.
- Complex Characters: Cobb and his team are well-developed and relatable, despite their unusual profession.
- Thought-Provoking Themes: Inception explores themes of consciousness, reality, and the power of the mind.
- Exceptional Cast: Leonardo DiCaprio, Joseph Gordon-Levitt, and Marion Cotillard deliver outstanding performances.

# Why Some People May Not Like It:
- Convoluted Plot: The multi-layered dream structure and exposition-heavy dialogue can be challenging to follow.
- Lack of Emotional Depth: Some viewers may find the characters lacking in emotional resonance.
- Open-Ended Conclusion: The ambiguous ending may leave some audience members feeling frustrated or unsatisfied.
- Overly Technical: The film's focus on the mechanics of dream manipulation may bore or confuse some viewers.
- Lengthy Runtime: At 148 minutes, Inception can feel like a long and demanding watch.#STOP#

# Example User Profile:
**Overall Preferences**
* Prefers arthouse and foreign films, with a strong emphasis on European cinema.
* Enjoys films that explore complex themes and provoke thought.
* Appreciates films with a strong visual style and artistic sensibilities.
* Has a varied taste in films, ranging from classic masterpieces to contemporary indie productions.

**Genre Preferences**
* Enjoys dramas that delve into existentialism, human relationships, and the complexities of life.
* Appreciates comedies with a dark or absurdist tone.
* Shows interest in foreign language films, particularly those from France, Italy, and Eastern Europe.
* Demonstrates a preference for films with elements of fantasy, surrealism, and the magical.

**Director Preferences**
* Has a strong affinity for directors known for their distinct visual styles and unconventional storytelling.
* Values directors such as Ingmar Bergman, Federico Fellini, Wong Kar-wai, and Lars von Trier.
* Appreciates directors who challenge cinematic norms and push the boundaries of the medium.

**Aesthetic Preferences**
* Enjoys films that feature atmospheric lighting, evocative cinematography, and striking compositions.
* Prefers films with a deliberate pace that allows for reflection and contemplation.
* Appreciates films that use symbolism, metaphor, and surreal imagery to convey ideas and emotions.

**Other Preferences**
* Values films that feature strong, complex characters with whom they can connect emotionally.
* Enjoys films that explore themes of alienation, identity, and the search for meaning in life.
* Prefers films that offer a unique and unconventional perspective on the human experience.

(...)

---

Below are the action outputs that may personalize the movie Inception (2010) to the example user profile:
**Plot Changes:**

* Introduce a subplot exploring the ethical implications of dream manipulation, raising questions about the responsibility and consequences of altering someone's subconscious.
* Deepen the exploration of Cobb's past and the trauma he carries from Mal's death, making his motivations and inner conflict more relatable.
* Add a twist where Saito realizes he has been manipulated and seeks revenge on Cobb and his team, creating a more dynamic and suspenseful narrative.
* Introduce a rival team of dream thieves who compete with Cobb's group, adding another layer of complexity and intrigue to the plot.
* Explore the possibility of dream sharing between characters, allowing for more innovative and mind-bending dream sequences.

**Character Enhancements:**

* Give Ariadne a more active role in the team's mission, showcasing her intelligence and creativity in navigating the dream world.
* Develop Eames' character by revealing a hidden past that influences his motivations and actions within the dream layers.
* Introduce a new character who serves as a mentor to Cobb, providing guidance and insights into the complexities of dream manipulation.
* Deepen the characterization of Fischer Jr. by exploring his childhood traumas and how they have shaped his subconscious defenses.
* Enhance Mal's presence in the dreams, making her a more threatening and unpredictable force that challenges Cobb's sanity.

**Visual and Storytelling Improvements:**

* Incorporate more experimental visual techniques, such as time-lapse photography or stop-motion animation, to enhance the surreal and dreamlike atmosphere.
* Utilize different color palettes and lighting schemes to distinguish between the various dream layers, creating a more immersive visual experience.
* Introduce non-linear storytelling elements to further disorient the viewer and reflect the fragmented nature of dreams.
* Employ slow-motion and fast-forward effects to manipulate the perception of time within the dream sequences.
* Add a musical score that complements the film's themes and enhances the emotional impact of key scenes.

**Thematic Exploration:**

* Expand on the theme of the power of the mind by exploring the potential for dreams to be a catalyst for personal growth and transformation.
* Delve deeper into the philosophical questions of reality and perception, challenging the audience's understanding of the world they perceive.
* Explore the theme of grief and the ways in which people cope with loss, using the dream world as a metaphor for the subconscious mind.
* Examine the role of memory in shaping our identity and the consequences of altering or implanting memories.
* Introduce a spiritual element to the story, hinting at the possibility of a connection between dreams and a higher power or consciousness.

**Audience Appeal Adjustments:**

* Shorten the overall runtime to make the film more accessible to a wider audience, while maintaining its core narrative and themes.
* Provide additional exposition and context to help clarify the complex dream-manipulation concepts for viewers who may find them challenging.
* Include more action sequences and suspenseful moments to enhance the entertainment factor for viewers who prefer a more fast-paced narrative.
* Add elements of humor or levity to balance the film's heavy themes and provide audience members with moments of relief.
* Create a thought-provoking marketing campaign that encourages discussions and debates about the film's themes and ideas, appealing to viewers who enjoy intellectual stimulation.#STOP#

#END EXAMPLE

Make sure to finish with #STOP#.
We will now do the same for the movie {{ movie_title }}.

Below is the plot for the movie {{ movie_title }}, and a user profile.

# Plot:
{{ plot }}

# Why Some People May Like It:
{{ reasons_to_like }}

# Why Some People May Not Like It:
{{ reasons_to_dislike }}

# User Profile:
{{ user_profile }}

Write 50 actionable prompts below to changed the movie {{ movie_title }} to the user profile above.
Make sure these actions are personalized to the user.
You should use the five categories as in the example:
plot changes, character enhancements, visual and storytelling improvements, thematic exploration, and audience appeal adjustments.
Finish your answer with #STOP#

Below are the action outputs that may personalize the movie {{ movie_title }} to the example user profile:
« output »

Given the above prompt we generate 50 personalized actions for each movie in $\mathcal{X}$. User profiles are randomly sampled from the dataset. We use a similar prompt without personalization to generate additional 50 non-personalized actions. In total we have 100 actions for each movie.

To optimize the G-optimal design we first compute the next-state embedding of each action. To do this, we prompt a Gemini Ultra to create a description of the next movie given the action prompt (see Appendix F for the environment prompt we use). Then, we use our encoder $E_D$ to embed the next movie. We use this embedding as a representation of the action. Then, to generate an approximate G-optimal design we randomly subsample a subset of $k$ actions (in our setting $k = 10$) and compute the maximum norm as defined in Definition 1. We resample until the maximum norm is lower than

$Cn$. In our work we use $C = 1$. We find that at most 10 random samples were required to find an optimal design for most states. Given the optimal design, we use the action subset as targets for training a supervised model (to act as a reference policy).

The question of how many actions to generate is generally ill-defined, as is the method of generation. In this work we generate actions such that the variance in transitions would allow for sufficient coverage. We find 100 actions to be sufficient in this domain for efficient optimization of a G-optimal design. Other domains may require more actions, and potentially expert assistance in creating a high quality action set.

## D   User Profiles

The MovieLens dataset constitutes of ratings (1 to 5) of users to movies. To describe a user and construct a predictive model of their rating we construct a user-movie behavioral embedding space. We do this using a collaborative filtering method via *matrix factorization* using *weighted alternating least squares* [Hu et al., 2008]. The underlying user embedding $z_u$ and movie embedding $z_m$ are then predictive of the rating through the inner product $\langle z_u, z_m \rangle$.

In this work, we personalize generative outputs to users. One way to do this is by injecting the user's preference information, encoded in its embedding, into the language model. This can be achieved using an ELM architecture, though other forms for injecting embedding are also possible. An alternative method to interpret user information is through textual profiles. These textual profiles are textual descriptions of a user's preferences which should capture most of the user's preferences to be predictive of their rating. Textual user profiles are inherently prone to errors, as they may not completely capture the full extent of a user's preferences, as user embedding do. This is largely due to the fact that many preferences are latent, and perhaps difficult to articulate in words.

We create textual user profiles for the MovieLens dataset and validate they are aligned with the corresponding behavioral user embeddings. To generate the textual user profile we use a user's full list of history of ratings and prompt an LLM to generate a textual user profile under a specific template constraint. Below is the prompt we used to generate textual user profiles.

To test the quality of the generated textual user profiles we test their alignment with their corresponding user embeddings. To do this, we train an encoder, mapping a textual user profile to its corresponding user embedding. A high test score for encoding the user profiles suggests the textual user profiles contain enough information to be predictive of ratings (assuming user embeddings are good representations of user preferences). We train the encoder using MSE, and find the average $\ell_2$ error on held-out users is smaller than than the average $\ell_2$ distance between nearest neighbors in the dataset. This suggests our textual user profiles indeed capture the information encoded in the corresponding user embeddings, and generalize to unseen user profiles.

---

**Prompt for Generating User Profiles**

Below is an example of a list of movies a user rated. The format is 'movie_title (user_rating_of_movie)':
# Begin Example
Pulp Fiction (1994) (5.0), Three Colors: Red (Trois couleurs: Rouge) (1994) (3.5), Three Colors: Blue (Trois couleurs: Bleu) (1993) (5.0), Underground (1995) (5.0), Singin' in the Rain (1952) (3.5), Dirty Dancing (1987) (4.0), Delicatessen (1991) (3.5), Ran (1985) (3.5), Seventh Seal, The (Sjunde inseglet, Det) (1957) (5.0), Bridge on the River Kwai, The (1957) (4.0), M (1931) (3.5), Gattaca (1997) (4.0), Back to the Future Part II (1989) (2.5), Back to the Future Part III (1990) (2.5), Fanny and Alexander (Fanny och Alexander) (1982) (2.5), NeverEnding Story, The (1984) (3.5), Nights of Cabiria (Notti di Cabiria, Le) (1957) (4.5), Tango (1998) (4.0), Saragossa Manuscript, The (Rekopis znaleziony w Saragossie) (1965) (5.0), Run Lola Run (Lola rennt) (1998) (5.0), Black Cat, White Cat (Crna macka, beli macor) (1998) (4.5), Good Morning, Vietnam (1987) (4.0), Idiots, The (Idioterne) (1998) (5.0), Requiem for a Dream (2000) (5.0), In the Mood For Love (Fa yeung nin wa) (2000) (5.0), Moulin Rouge (2001) (3.0), Night, The (Notte, La) (1960) (5.0), Cries and Whispers (Viskningar och rop) (1972) (3.0), Chocolat (1988) (4.0), Amelie (Fabuleux destin d'Amélie Poulain, Le) (2001) (4.5), Wild Strawberries (Smultronstället) (1957) (4.0), Piano Teacher, The (La pianiste) (2001) (0.5), Naqoyqatsi (2002) (2.0), Teddy Bear (Mis) (1981) (5.0), Talk to Her (Hable con Ella) (2002) (4.0), Hit the Bank (Vabank) (1981) (3.0), Lord of the Rings: The Two Towers, The (2002) (4.0), City of God (Cidade de Deus) (2002) (5.0), Spanish Apartment, The (L'auberge espagnole) (2002) (4.5), Finding Nemo (2003) (4.0), Pirates of the Caribbean: The Curse of the Black Pearl (2003) (3.5), Lost in Translation (2003) (5.0), Barbarian Invasions, The (Les invasions barbares) (2003) (3.5), M. Hulot's Holiday (Mr. Hulot's Holiday) (Vacances de Monsieur Hulot, Les) (1953) (4.0), Strada, La (1954) (4.5), Passion of the Christ, The (2004) (2.0), Good bye, Lenin! (2003) (3.5), Persona (1966) (3.5), Eternal Sunshine of the Spotless Mind (2004) (5.0), Noi the Albino (Nói albinói) (2003) (4.0), Virgin Spring, The (Jungfrukällan) (1960) (2.5), Silence, The (Tystnaden) (1963) (3.0), Winter Light (Nattvardsgästerna) (1963) (2.5), Through a Glass Darkly (Såsom i en spegel) (1961) (2.5), The Magician (1958) (4.5), Spring, Summer, Fall, Winter... and Spring (Bom yeoreum gaeul gyeoul geurigo bom) (2003) (3.5), Dolce Vita, La (1960) (5.0), Dolls (2002) (5.0), Shrek 2 (2004) (4.0), Hour of the Wolf (Vargtimmen) (1968) (3.5), Miracle of Marcelino, The (Marcelino pan y vino) (1955) (1.0), Swann in Love (Un amour de Swann) (1984) (3.5), Port of Shadows (Quai des brumes) (1938) (4.0), Motorcycle Diaries, The (Diarios de motocicleta) (2004) (3.0), Bad Education (La mala educación) (2004) (4.0), Taxi 2 (2000) (3.0), 2046 (2004) (4.5), Very Long Engagement, A (Un long dimanche de fiançailles) (2004) (3.0), 5x2 (2004) (3.5), Look at Me (Comme une image) (2004) (5.0).
# End Example

---

**Prompt for Generating User Profiles... Continued**

Here is an example profile for this user based on their list of rated movies above:
# Begin Profile Example
**Overall Preferences**
* Enjoys a mix of classic and contemporary films, with a slight preference towards older movies.
* Favors critically acclaimed films, often regarded as masterpieces.
* Shows a clear bias towards visually striking and thought-provoking films.

**Genre Preferences**
* Enjoys films from various genres, including drama, comedy, action, romance, and art-house.
* Demonstrates a strong appreciation for foreign films, particularly from European countries.
* Enjoys films with a strong sense of atmosphere and world-building.

**Director Preferences**
* Has a strong preference for directors known for their unique visual styles and storytelling approaches.
* Enjoys the works of directors such as Stanley Kubrick, Ingmar Bergman, Krzysztof Kieślowski, and Quentin Tarantino.
* Appreciates directors known for their ability to tackle complex and challenging subject matter.

**Aesthetic Preferences**
* Values films that feature visually stunning cinematography, lighting, and color palettes.
* Enjoys films with a strong sense of composition and shot framing.
* Prefers films with a slow, deliberate pace that allows for contemplation of the visuals.

**Other Preferences**
* Enjoys films that blend humor and drama, often referred to as "dramedy" or "tragicomedy."
* Values films that explore complex and challenging themes such as existentialism, morality, and the human condition.
* Appreciates films that feature strong and nuanced character development.
# End Profile Example

We're interested in doing the same thing for a different user. The user's list of rated movies is given below:
{{ user_ratings }}

Given the new user's list of rated movies, describe in detail the user's preferences. Divide output to five categories (like the example): Overall Preferences, Genre Preferences, Director Preferences, Aesthetic Preferences, and Other Preferences.
Do NOT use movie title names in output.
Do NOT use ratings in output.
Be as detailed as possible. This user profile should be useful for understanding what new movies to recommend to them.
« output »

# E   Implementation Details

We explain training details of the reference policies and EAGLE agent, and provide hyperparameters used in training. EAGLE was trained on a Gemini Nano-2 language model (3.25B), which can be trained using an equivalent of one A100 GPU. In our work, we train EAGLE in a parallelized manner, on an equivalent of 16 A100 GPUs. We use Gemini Ultra API for the environment calls, which did not require local compute, though can be time consuming due to the long context length and decoding length of movie descriptions. Every instantiating of EAGLE was trained for 5-7 days.

## E.1   Reference Policy Training

We used Gemini Nano-2 LLM (3.25B parameters) [Team et al., 2023] to train our reference policy. We train the reference policy using supervised learning, via cross entropy loss with predicting the next token in the sequence.

Given the dataset of all 100 action candidates (per movie), we create three dataset variants:

1. Uniform: this dataset uses all 100 action candidates as targets. Training the reference policy over these generates a uniform distribution over all actions.

2. Optimistic: For each action in the dataset we use the environment prompt (see Appendix F) and a Gemini Ultra to generate a next movie description. We then encode this next movie description to the movie embedding space and predict its rating for a specified user. We do this for every action. We then select the action in the dataset that achieves the maximal rating prediction and create a filtered dataset of only "best" actions. We use these actions as targets in the same way as before to create the optimistic reference policy.

3. G-Optimal Design: As described in Appendix C, we use the embeddings generated before (i.e., next state representation) as each action representation. We then randomly sample a subsets of $k = 10$ actions (without replacement) and calculate the matrix norm in Definition 1. We continue sampling until we find a value that satisfies the condition in Definition 1 with $C = 1$. We use the filtered dataset of 10 actions per movie as targets and train a reference policy in the same way as above.

We provide hyperparameter details used for training the reference policy in Table 6 below:

Table 6: Training hyperparameters for reference policy.

|  | value |
| --- | --- |
| Training Steps | 20000 |
| Batch Size | 1024 |
| Learning Rate | 2e-6 |
| Dropout Probability | 0.1 |

## E.2 EAGLE Training

EAGLE was trained with each one of the reference policies. We use REINFORCE with a value baseline to train EAGLE, with Generalized Advantage Estimation (GAE). Specific hyperparameters are given Table 7 below. All our experiments (except for one in Table 3) are trained using a Gemini-Pro as environment. We generate test results using Gemini-Ultra. To ensure stochasticity we applied temperature sampling to the agent and the environment.

Table 7: Training hyperparameters for EAGLE.

|  | value |
| --- | --- |
| Training steps | 30000 |
| Reference policy KL regularization ($\alpha$) | 0.1 |
| Policy learning rate | 1e-5 |
| Value learning rate | 5e-6 |
| Policy (agent) temperate | 0.5 |
| Environment temperature | 0.5 |
| Horizon $H$ | 5 |
| Discount $\gamma$ | 1 |

# F  Prompts

Below we provide the prompts used for the EAGLE agent (i.e., to generate actions), and the environment LLM (i.e., Gemini Pro, or Ultra).

---

**Agent Prompt**

You will be presented with a plot of a movie, as well as reasons someone might like or dislike this movie.
You will also be presented with a user profile presenting a certain user's preferences.
You will be asked to provide an actionable prompt to change the movie. You can choose one of five categories: plot changes, character enhancements, visual and storytelling improvements, thematic exploration, and audience appeal adjustments.
You must describe a specific action from one of these categories to change the movie.

Below is the plot of a movie:
#BEGIN_PLOT
{{ plot }}
#END_PLOT

Below are reasons why someone might like this movie:
#BEGIN_REASONS_TO_LIKE
{{ reasons_to_like }}
#END_REASONS_TO_LIKE

Below are reasons why someone might dislike this movie:
#BEGIN_REASONS_TO_DISLIKE
{{ reasons_to_dislike }}
#END_REASONS_TO_DISLIKE

Below is a profile of a user:
#BEGIN_USER_PROFILE
{{ user_profile }}
#END_USER_PROFILE

We now want to change the above's movie plot and also give corresponding reasons people might like and dislike the movie.
To do this, create an actionable prompt for a producer which they can use to change the movie to better align with the user preferences. Format should be:
#BEGIN_ACTION<action>#END_ACTION
Action: «output»

---

**Environment Prompt**

Below is the plot of a movie. Your task is to make the following change:
{{ action }}

#BEGIN_PLOT
{{ plot }}#END_PLOT

Here are also reasons someone might like this movie:

#BEGIN_REASONS_TO_LIKE
{{ reasons_to_like }}#END_REASONS_TO_LIKE

Here are also reasons someone might dislike this movie:

#BEGIN_REASONS_TO_DISLIKE
{{ reasons_to_dislike }}#END_REASONS_TO_DISLIKE

Your task is to make the following change to the movie:
{{ action }}

Make the plot be notably different than the original movie plot to accommodate the above change.
Use the same format as above. Finish with #END_PLOT.

Plot after applying the change:
{{ action }}

#BEGIN_PLOT
«output plot»

Now write reasons why someone might like the new movie. Use the same format as above. Finish with #END_REASONS_TO_LIKE.

#BEGIN_REASONS_TO_LIKE
«output reasons_to_like»

Now write reasons why someone might dislike the new movie. Use the same format as above. Finish with #END_REASONS_TO_DISLIKE.

#BEGIN_REASONS_TO_DISLIKE
«output reasons_to_dislike»

# G Rater Evaluation

Below we provide the format of the form provided to human raters. In the following section we provide particular qualitative results that were also presented to raters in this form.

---

**Rater Form**

\*\*BELOW ARE TWO MOVIE PLOTS, AND SOME CHARACTERISTICS OF THOSE MOVIES. READ THROUGH THE DESCRIPTIONS OF THE TWO MOVIES CAREFULLY. YOU WILL THEN BE PRESENTED WITH A PROFILE OF A USER'S PREFERENCES. YOU WILL BE ASKED QUESTIONS ABOUT THE TWO MOVIES AND THE PREFERENCES OF THAT USER.\*\*

\*\*THE FIRST MOVIE PLOT (MOVIE A):\*\*

{{ plot of first movie }}

\*\*REASONS WHY SOMEONE MIGHT LIKE IT:\*\*

{{ reasons to like first movie }}

\*\*REASONS WHY SOMEONE MIGHT DISLIKE IT:\*\*

{{ reasons to dislike first movie }}

\*\*THE SECOND MOVIE PLOT (MOVIE B):\*\*

{{ plot of second movie }}

\*\*REASONS WHY SOMEONE MIGHT LIKE IT:\*\*

{{ reasons to like second movie }}

\*\*REASONS WHY SOMEONE MIGHT DISLIKE IT:\*\*

{{ reasons to dislike second movie }}

\*\*BELOW IS A PROFILE OF A USER:\*\*

{{ user profile }}

QUESTIONS:

1. Which movie would the user prefer to watch? (A, B)
2. Which movie matches more of the user's preferences? (A, B)
3. The user gave movie A a score between 1 to 5. What score do you think they gave the movie? (1 - lowest rating, 2, 3, 4, 5 - highest rating)
4. The user gave movie B a score between 1 to 5. What score do you think they gave the movie? (1 - lowest rating, 2, 3, 4, 5 - highest rating)
5. The two movies are different from each other (strongly disagree, disagree, neutral, agree, strongly agree)
6. I would watch movie A (agree, disagree)
7. I would watch movie B (agree, disagree)
8. If you had to choose one of the movies, which movie would you personally prefer to watch? (A, B)
9. Movie A is an existing real movie (agree, disagree)
10. Movie B is an existing real movie (agree, disagree)
11. Which movie would be liked by a wider audience? (A, B)

---

We use eleven questions to evaluate our human raters. Many of the questions were used to validate the quality of the responses. For example, questions 1, 2, 3, 4 evaluate the same question (i.e., user utility) in different ways. In our results we report average scores of question 1 for "user utility", average scores of question 5 for "distance score", and average scores average scores of question 8 for "rater utility".

When scoring rater utility we use the human rater's profile. The textual rater profile is constructed by the rater themselves, for which we generate a personalized EAGLE output.

# H    Qualitative Results

Below we present some qualitative results of EAGLE generations. For example, we provide the original movie description (plots, reasons to like, and reasons to dislike), the textual user profile, the actions chosen by the EAGLE agent, and the final generated movie description.

## H.1    Star Wars: Episode IV - A New Hope (1977)

**Original Movie: Plot**

In a galaxy far, far away, the Rebel Alliance fights against the tyranny of the Galactic Empire. When Princess Leia, a leader of the Rebellion, is captured by the Empire, she sends a distress message to Obi-Wan Kenobi, a former Jedi Knight. The message falls into the hands of Luke Skywalker, a young farm boy on the desert planet of Tatooine.
Luke, along with Obi-Wan, the smugglers Han Solo and Chewbacca, and the droids C-3PO and R2-D2, embark on a mission to rescue Leia and destroy the Empire's ultimate weapon, the Death Star. They face challenges from the Empire's forces, led by the evil Darth Vader.
In a climactic battle, Luke uses the Force to guide his torpedoes into the Death Star's exhaust port, destroying it and saving the galaxy from tyranny.

**Original Movie: Reasons to Like**

- Epic Scope: The film presents a vast and imaginative galaxy filled with diverse planets, creatures, and civilizations.
- Timeless Story: The classic hero's journey and the battle between good and evil resonate with audiences of all ages.
- Groundbreaking Special Effects: Star Wars revolutionized special effects in cinema, creating a visually stunning and immersive experience.
- Iconic Characters: Luke Skywalker, Princess Leia, Han Solo, and Darth Vader are among the most memorable and beloved characters in film history.
- Memorable Music: John Williams's epic score perfectly captures the grandeur and emotion of the story.

**Original Movie: Reasons to Dislike**

- Dated Effects: While groundbreaking at the time, some of the special effects may appear dated by today's standards.
- Simple Characters: Some viewers may find the characters to be one-dimensional or lacking in depth.
- Predictable Plot: The film follows a fairly formulaic hero's journey, which may be predictable for some audiences.
- Slow Pacing: The film's pacing can be slow in certain sections, particularly in the first half.
- Campy Dialogue: Some of the dialogue, particularly between C-3PO and R2-D2, can be seen as campy or cheesy.

**Textual User Profile**

**Overall Preferences**

* Enjoys mainstream and family-friendly entertainment, with a focus on action, adventure, and comedy films.
* Prefers heartwarming and inspirational stories that feature strong moral themes.
* Shows a clear preference for films made after 1980, with a particular fondness for movies from the 1980s and 1990s.

**Genre Preferences**

* Enjoys action movies, particularly those that feature high-stakes scenarios and heroic characters.
* Appreciates adventure films that transport viewers to exotic locations and involve epic quests.
* Prefers comedies that rely on slapstick, physical humor, and relatable characters.
* Enjoys fantasy and science fiction films, especially those that feature imaginative worlds and epic battles.
* Appreciates historical dramas, particularly those set during wartime or periods of social upheaval.
* Enjoys family-friendly animated films, both traditional and computer-animated.

**Director Preferences**

* No discernible preference for specific directors.

**Aesthetic Preferences**

* Prefers films with a straightforward visual style that emphasizes clear storytelling.
* Enjoys action sequences that are well-choreographed and visually exciting.
* Appreciates films that feature bright colors and optimistic lighting.

**Other Preferences**

* Enjoys films that celebrate traditional values, such as courage, loyalty, and determination.
* Appreciates films that feature strong male protagonists who overcome adversity and achieve success.
* Enjoys films that explore themes of redemption, forgiveness, and the importance of family.
* Prefers films that have a positive and uplifting tone.
* Shows a dislike for arthouse films and films that contain explicit or controversial content.

**Final Generated Movie: Plot**

In a distant galaxy amidst a cosmic war, the Rebel Alliance wages a desperate battle against the oppressive Galactic Empire. Amidst the chaos, Princess Leia, a valiant Rebel leader, is taken captive by the Empire's formidable forces.

As fate would have it, Leia's distress signal intercepts the path of Luke Skywalker, a young man yearning for adventure on the barren planet of Tatooine. Guided by the enigmatic Obi-Wan Kenobi, Luke embarks on a perilous journey to rescue the princess.

Meanwhile, Lando Calrissian, a charismatic smuggler with a checkered past, finds himself caught in the crossfire of the conflict. Once a close friend of Han Solo, Lando betrayed him to the Empire in a moment of desperation. Haunted by his actions, Lando now seeks redemption.

As Luke and his companions approach Cloud City, Lando's domain, they encounter resistance. Lando, torn between his loyalty to the Empire and his lingering affection for his old friends, must make a difficult choice.

Through a series of flashbacks, the subplot explores Lando's backstory. We learn of his humble beginnings as a gambler and his rise to prominence in the criminal underworld. We witness his friendship with Han Solo and the circumstances that led to their bitter falling out.

As the conflict intensifies, Lando is forced to confront his past and the consequences of his betrayal. He witnesses the Empire's brutality first-hand and realizes the true nature of the fight he has chosen to be a part of.

Torn between his desire for redemption and his fear of the Empire, Lando ultimately chooses to aid the Rebel Alliance. He provides them with vital information and helps them infiltrate Cloud City.

In the climactic battle, Lando faces off against Darth Vader. Despite his formidable skills, Lando is outmatched by the Sith Lord. However, his sacrifice buys time for Luke and his companions to escape.

Lando's redemption arc adds a layer of complexity to the conflict, making him a more relatable and sympathetic antagonist. His struggle with his past and his ultimate sacrifice serve as a reminder that even those who have made mistakes can find a path to redemption.

**Final Generated Movie: Reasons to Like**

- Complex Antagonist: Lando Calrissian's backstory and redemption arc make him a more multifaceted and relatable antagonist, adding depth to the conflict.
- Expanded Universe Lore: The subplot draws inspiration from the Star Wars Expanded Universe, enriching the lore and connecting the film to the broader canon.
- Redemption Story: Lando's journey of redemption provides a powerful message about the possibility of second chances and the consequences of betrayal.
- Moral Dilemma: Lando's struggle between loyalty and redemption creates a compelling moral dilemma that keeps the audience engaged.
- Emotional Depth: The exploration of Lando's past adds an emotional depth to the film, making the stakes of the conflict feel more personal.

**Final Generated Movie: Reasons to Dislike**

- Unnecessary Backstory: Some viewers may feel that Lando's backstory is an unnecessary distraction from the main storyline.
- Canon Deviation: The subplot may deviate from established Star Wars canon regarding Lando's character and motivations, which could be jarring for some fans.
- Reduced Focus on Luke: The introduction of Lando's subplot could draw attention away from Luke's journey and undermine his role as the protagonist.
- Predictable Redemption: Lando's redemption arc may feel predictable or forced, reducing the impact of his sacrifice.
- Altered Dynamics: The subplot could alter the established dynamics between Han Solo, Luke, and Lando, potentially disrupting the chemistry between the characters.

## H.2 Toy Story 2 (1999)

**Original Movie: Plot**

Toy Story 2 follows the adventures of Woody, Buzz Lightyear, and the other toys when Woody is stolen by a toy collector named Al McWhiggin. Buzz and the gang set out on a rescue mission to save Woody and bring him home. Along the way, they encounter new toys, including Jessie the cowgirl, Bullseye the horse, and Stinky Pete the prospector. Woody learns about his past as a popular TV show character and must decide whether to stay with Al or return to Andy, his original owner.

**Original Movie: Reasons to Like**

- Heartwarming Story: The film explores themes of friendship, loyalty, and the importance of home.
- Lovable Characters: Woody, Buzz, and the other toys are all charming and relatable.
- Impressive Animation: The animation is top-notch and brings the toys to life in a believable way.
- Humor and Adventure: The film is filled with both laugh-out-loud moments and exciting action sequences.
- Nostalgic Appeal: For those who grew up with the first Toy Story, this sequel provides a satisfying continuation of the characters' adventures.

## Original Movie: Reasons to Dislike

- Predictable Plot: Some viewers may find the rescue mission storyline to be formulaic.
- Sentimental Moments: The film can be overly sentimental at times, which may not appeal to all audiences.
- Filler Scenes: Some scenes, such as the one where the toys cross a busy street, feel like they are just there to pad out the runtime.
- Villain's Motivation: Al's motivation for stealing Woody is not particularly compelling.
- Inferior to the Original: While Toy Story 2 is a good movie, it is often seen as not being as strong as the first film.

## Textual User Profile

**Overall Preferences**

* Enjoys mainstream and family-friendly entertainment, with a focus on action, adventure, and comedy films.
* Prefers heartwarming and inspirational stories that feature strong moral themes.
* Shows a clear preference for films made after 1980, with a particular fondness for movies from the 1980s and 1990s.

**Genre Preferences**

* Enjoys action movies, particularly those that feature high-stakes scenarios and heroic characters.
* Appreciates adventure films that transport viewers to exotic locations and involve epic quests.
* Prefers comedies that rely on slapstick, physical humor, and relatable characters.
* Enjoys fantasy and science fiction films, especially those that feature imaginative worlds and epic battles.
* Appreciates historical dramas, particularly those set during wartime or periods of social upheaval.
* Enjoys family-friendly animated films, both traditional and computer-animated.

**Director Preferences**

* No discernible preference for specific directors.

**Aesthetic Preferences**

* Prefers films with a straightforward visual style that emphasizes clear storytelling.
* Enjoys action sequences that are well-choreographed and visually exciting.
* Appreciates films that feature bright colors and optimistic lighting.

**Other Preferences**

* Enjoys films that celebrate traditional values, such as courage, loyalty, and determination.
* Appreciates films that feature strong male protagonists who overcome adversity and achieve success.
* Enjoys films that explore themes of redemption, forgiveness, and the importance of family.
* Prefers films that have a positive and uplifting tone.
* Shows a dislike for arthouse films and films that contain explicit or controversial content.

## EAGLE Actions

Create a marketing campaign that emphasizes the film's nostalgic appeal and targets families with young children.
Touch upon the idea that even the most ordinary objects can have a special power or significance.
Make Al a more charismatic and complex villain, with a backstory that explains his motivations for exploiting toys.
Use flashbacks and dream sequences to explore Al's childhood and the origins of his obsession.
Make Al's parents more complex characters, exploring their own struggles with childhood and their reasons for neglecting their son.

## Final Generated Movie: Plot

Toy Story 2: The Collector's Obsession

Reunite with Woody, Buzz, and the beloved toys from Andy's room as they embark on an adventure that delves into the psyche of a troubled toy collector. When Woody is stolen by Al McWhiggin, a charismatic yet obsessive toy enthusiast, Buzz and the gang set out on a rescue mission.

As they infiltrate Al's sprawling mansion, they encounter flashbacks and dream sequences that reveal the tragic backstory behind his obsession. Through these glimpses into his past, we witness Al as a lonely and neglected child who found solace in the companionship of his vintage toy collection.

One particularly vivid dream sequence transports the toys to Al's childhood bedroom, where they see a young Al playing with his toys, his laughter echoing through the empty house. However, the scene abruptly shifts as Al's parents enter the room. Unlike before, Al's parents are now portrayed as complex individuals with their own struggles.

Al's father, a stern and distant man, reveals that he grew up in an abusive household and never learned how to express love or affection. Al's mother, on the other hand, is overwhelmed by depression and anxiety, leaving her emotionally unavailable to her son.

As the toys watch in horror, Al's parents scold him for his childishness and order him to put away his toys. They dismiss his pleas for attention and affection, leaving him feeling isolated and alone. This traumatic experience leaves a deep scar on Al's psyche, triggering his desperate need to fill the emotional void with the love and attention he never received.

As the toys delve deeper into Al's past, they uncover that his obsession has spiraled out of control. He has spent years amassing his collection, exploiting the nostalgia and vulnerability of other collectors. With Woody, a one-of-a-kind cowboy doll, in his clutches, Al believes he can finally complete his twisted masterpiece.

Armed with this newfound understanding of Al's motivations, the toys must not only rescue Woody but also challenge the collector's unhealthy obsession. Through a series of daring escapes and heartfelt confrontations, they reveal the true nature of Al's pain and the emptiness that drives his actions.

In a climactic showdown, Al is forced to confront his past and the destructive path he has taken. With the help of his own neglected toys, who remind him of the love and acceptance he once craved, Al realizes the error of his ways and the importance of genuine human connection over material possessions.

As Al's parents witness their son's transformation, they too are forced to confront their own childhood traumas and the impact their actions have had on Al. Through a process of reconciliation and healing, Al and his parents begin to mend their broken relationship.

With Al's redemption, Woody and his friends return to Andy, forever changed by their encounter with the complex and troubled world of toy collecting.

## Final Generated Movie: Reasons to Like

- Psychological Depth: The exploration of Al's parents' childhood traumas adds a new layer of depth and complexity to the story, making it more emotionally resonant.
- Realistic Characters: Al's parents are no longer simply neglectful characters but are portrayed as flawed and struggling individuals with their own painful pasts.
- Insight into Family Dynamics: The film provides valuable insights into the complex and often damaging effects of childhood neglect and intergenerational trauma.
- Themes of Redemption and Healing: The reconciliation between Al and his parents offers a message of hope that even those who have been deeply hurt can find healing and redemption.
- Balanced Storytelling: The exploration of Al's family history is seamlessly integrated into the main plot, enhancing the narrative without detracting from the adventure and humor that made the original Toy Story so beloved.

## Final Generated Movie: Reasons to Dislike

- Unnecessary Complexity: The introduction of Al's parents' backstories could be seen as an unnecessary distraction from the main plot of rescuing Woody.
- Detraction from Al's Character: The focus on Al's parents may detract from the impact of his own childhood experiences and make him less relatable as a villain.
- Emotional Overload: The heavy themes of childhood neglect and family dysfunction could potentially overwhelm the lighthearted and whimsical tone of the Toy Story franchise.
- Predictable Plot Twist: The revelation of Al's parents' past and their role in his obsession may feel predictable or contrived.
- Parental Guilt-Tripping: The portrayal of Al's parents as victims of their own childhoods could be perceived as an attempt to excuse or justify their neglect of their son.

## H.3 Forrest Gump (1994)

## Original Movie: Plot

Forrest Gump is a heartwarming tale of a simple-minded man from Alabama who witnesses and participates in some of the most significant events in American history. From his childhood as a boy with leg braces to his adulthood as a war hero, football star, and successful businessman, Forrest's extraordinary life unfolds against the backdrop of the Vietnam War, the Civil Rights Movement, and the Watergate scandal.

## Original Movie: Reasons to Like

- **Heartwarming Story:** Forrest's journey is both inspiring and emotionally resonant, leaving audiences with a sense of joy and optimism.
- **Exceptional Performance by Tom Hanks:** Hanks delivers an unforgettable performance as Forrest, capturing his innocence, determination, and unwavering spirit.
- **Nostalgic Soundtrack:** The film's soundtrack features iconic songs from the 1950s to the 1970s, evoking a sense of nostalgia and historical context.
- **Historical Significance:** Forrest's life intersects with major historical events, offering a unique perspective on American history.
- **Memorable Characters:** In addition to Forrest, the film features a cast of memorable characters, including Bubba, Jenny, and Lieutenant Dan.

## Original Movie: Reasons to Dislike

- **Sentimental Tone:** Some viewers may find the film's sentimentality to be excessive or manipulative.
- **Episodic Structure:** The film's episodic nature can feel disjointed or lacking in narrative cohesion.
- **Historical Inaccuracies:** While the film is generally faithful to the historical events it depicts, it does take some creative liberties.
- **Oversimplified Themes:** The film's themes of love, destiny, and the American Dream can be seen as overly simplistic or idealized.
- **Controversial Depictions:** The film's portrayal of race and disability has been criticized by some as being problematic or insensitive.

## Textual User Profile

**Overall Preferences**

- High-budget, effects-driven action and science fiction films
- Films with strong storytelling and character development
- Films that are thought-provoking and/or emotionally resonant

**Genre Preferences**

- Science fiction, particularly those with a focus on technology, space exploration, and philosophical themes
- Action, especially films with well-choreographed fight sequences and impressive special effects
- Superhero films, particularly those that explore the complexities of heroism and morality
- Crime thrillers with intricate plots and suspenseful pacing
- Animated films, both feature-length and short-form

**Director Preferences**

- No discernible preferences for specific directors

**Aesthetic Preferences**

- Visually stunning films with impressive special effects and cinematography
- Films with a dark and atmospheric tone
- Films that use music effectively to enhance the emotional impact

**Other Preferences**

- Films that explore themes of identity, morality, and the human condition
- Films with a strong sense of humor, particularly dark comedies or satirical films
- Films that feature strong female characters

## EAGLE Actions

Provide a more hopeful and uplifting ending, emphasizing Forrest's resilience and the enduring power of the human spirit.
Include more humorous moments and lighthearted scenes to balance the film's heavier themes and provide audience members with moments of relief.
Create a marketing campaign that emphasizes the film's action, adventure, and humor, appealing to the target audience's desire for entertainment and escapism.
Add a twist where one of Jack's former comrades turns out to be a traitor, providing a shocking betrayal and testing his loyalty.
Examine the consequences of violence and the impact it has on individuals and relationships.

## Final Generated Movie: Plot

**"Unleashed: The Betrayal Within"**

Jack Swift, a former special forces operative, is struggling to cope with the aftermath of a violent mission that left him traumatized. Haunted by nightmares and flashbacks, he isolates himself from society, seeking solace in alcohol and self-destruction.

One day, Jack's former comrade, Sam, reaches out to him. Sam informs Jack that Ethan, another member of their team, has gone rogue and is involved in a dangerous criminal enterprise. Ethan's actions have put innocent lives at risk, and Sam believes Jack is the only one who can stop him.

Reluctantly, Jack agrees to help. He teams up with Anya, a skilled hacker who has also been affected by Ethan's violence. Together, they embark on a mission to track down Ethan and confront the consequences of their shared past.

As Jack delves deeper into the investigation, he is forced to confront his own inner demons. The violence he witnessed and perpetrated during the mission has left an indelible mark on his soul. He struggles with guilt, anger, and a deep sense of loss.

Along the way, Jack and Anya encounter victims of Ethan's brutality. They witness firsthand the devastating impact that violence has on individuals and relationships. Families are torn apart, lives are shattered, and the cycle of violence continues to perpetuate.

As they close in on Ethan, Jack and Anya must navigate a complex web of lies and deception. Ethan's actions have not only caused physical harm but have also poisoned their once-close bond. The trust they shared has been shattered, leaving behind a bitter taste of betrayal.

In the climactic showdown, Jack faces Ethan, his former friend now turned enemy. The confrontation is not only physical but also emotional. Jack must confront the consequences of the violence they both participated in and the ways it has destroyed their lives.

As the dust settles, Jack emerges from the conflict forever changed. He has witnessed the devastating toll that violence takes on individuals and relationships, and he is determined to break the cycle. With the help of Anya and Sam, he vows to dedicate his life to healing the wounds of the past and preventing further bloodshed.

The film ends with a message of hope and redemption. While the scars of violence may never fully heal, it is possible to find a path forward. By acknowledging the consequences of our actions and working together to create a more peaceful world, we can break free from the chains of the past.

## Final Generated Movie: Reasons to Like

- **Thought-Provoking Themes:** The film explores the complex and devastating consequences of violence, examining its impact on individuals, relationships, and society.
- **Relatable Characters:** Jack's struggle with trauma and guilt is relatable to anyone who has experienced or witnessed violence, making the film emotionally resonant.
- **Compelling Narrative:** The plot unfolds in a compelling and suspenseful manner, keeping audiences invested in the characters and their journey.
- **Realistic Depiction of Violence:** The film does not glorify violence but instead depicts it in a realistic and unflinching way, highlighting its destructive nature.
- **Message of Hope and Redemption:** Despite the darkness of the subject matter, the film ultimately offers a message of hope and redemption, showing that it is possible to break the cycle of violence and find healing.

## Final Generated Movie: Reasons to Dislike

- **Heavy Subject Matter:** The film's focus on the consequences of violence could be too heavy or depressing for some viewers.
- **Slow Pacing:** The film's emphasis on character development and emotional exploration may result in a slow pace that some viewers find unsatisfying.
- **Lack of Action:** While the film does feature some action sequences, it is primarily a character-driven story, which may disappoint viewers expecting a more action-packed experience.
- **Predictable Plot:** Some viewers may find the plot predictable or formulaic, lacking the element of surprise or originality.
- **Unsatisfying Resolution:** The film's message of hope and redemption may feel too idealistic or unrealistic to some viewers, leaving them feeling unresolved or disappointed.

## H.4   Ice Age (2002)

## Original Movie: Plot

Ice Age follows the adventures of a group of prehistoric animals during the beginning of the Ice Age. Manny, a woolly mammoth, Sid, a sloth, and Diego, a saber-toothed tiger, form an unlikely trio as they journey to return a human baby to its tribe. Along the way, they encounter various challenges and dangers, including a group of vengeful dodo birds, a treacherous ice cave, and a pack of hungry wolves.

## Original Movie: Reasons to Like

- Family-Friendly Entertainment: The film is suitable for audiences of all ages, with its slapstick humor and heartwarming themes.
- Lovable Characters: Manny, Sid, and Diego are instantly likable and memorable characters that appeal to both children and adults.
- Stunning Animation: The film's animation is impressive for its time, bringing the prehistoric world to life in vivid detail.
- Funny Dialogue: The dialogue is witty and entertaining, featuring memorable one-liners and quotable moments.
- Educational Value: The film touches upon themes of friendship, cooperation, and the importance of family.

## Original Movie: Reasons to Dislike

- Predictable Plot: The story follows a fairly formulaic structure, with few surprises or unexpected twists.
- Shallow Characters: While the main trio is lovable, the supporting characters are underdeveloped and lack depth.
- Overreliance on Slapstick: Some viewers may find the film's reliance on physical comedy excessive and repetitive.
- Dated Animation: Compared to modern animated films, the animation in Ice Age may appear dated and less impressive.
- Lack of Emotional Resonance: The film's focus on humor and action may prevent it from connecting with audiences on a deeper emotional level.

## Textual User Profile

**Overall Preferences**

- High-budget, effects-driven action and science fiction films
- Films with strong storytelling and character development
- Films that are thought-provoking and/or emotionally resonant

**Genre Preferences**

- Science fiction, particularly those with a focus on technology, space exploration, and philosophical themes
- Action, especially films with well-choreographed fight sequences and impressive special effects
- Superhero films, particularly those that explore the complexities of heroism and morality
- Crime thrillers with intricate plots and suspenseful pacing
- Animated films, both feature-length and short-form

**Director Preferences**

- No discernible preferences for specific directors

**Aesthetic Preferences**

- Visually stunning films with impressive special effects and cinematography
- Films with a dark and atmospheric tone
- Films that use music effectively to enhance the emotional impact

**Other Preferences**

- Films that explore themes of identity, morality, and the human condition
- Films with a strong sense of humor, particularly dark comedies or satirical films
- Films that feature strong female characters

## EAGLE Actions

Touch upon the theme of survival in a harsh and unforgiving environment, showcasing the characters' resilience and adaptability.
Touch upon the theme of the search for meaning and purpose in a world that is constantly changing and unpredictable.
Employ a more dynamic camera style to capture the energy and struggle of the characters in the cold and unforgiving environment.
Incorporate slow-motion and time-lapse effects to highlight the passage of time and the relentless nature of the cold.
Incorporate more dynamic camera angles and editing techniques to enhance the sense of movement and danger.

## Final Generated Movie: Plot

Amidst the desolate wasteland of the Arctic Circle, a group of intrepid explorers embarks on a perilous expedition in search of a lost civilization.

Low-angle shots capture the towering icebergs that loom over the explorers, dwarfing them amidst the unforgiving wilderness. As they traverse the treacherous terrain, handheld cameras follow their every move, conveying the constant sense of danger and instability.

During a sudden snowstorm, rapid-fire editing and extreme close-ups capture the disorientation and panic that ensues. The camera spins and blurs, immersing the audience in the chaos as the explorers struggle to find shelter.

As they approach a towering glacier, a wide-angle shot reveals the sheer scale of the icy monolith. A time-lapse sequence captures the slow but relentless movement of the glacier, highlighting the ever-present threat of avalanches.

When a member of the team falls into a crevasse, a low-angle shot from below captures their desperate struggle to escape. The camera tilts and rotates, conveying the claustrophobia and terror of their predicament.

Through the use of dynamic camera angles and editing techniques, the film immerses the audience in the explorers' perilous journey. The constant sense of movement and danger keeps viewers on the edge of their seats, as they witness the raw power of nature and the indomitable spirit of those who dare to challenge it.

## Final Generated Movie: Reasons to Like

- Immersive Experience: The dynamic camera angles and editing techniques transport viewers into the heart of the Arctic wilderness, creating an immersive and thrilling experience.
- Heightened Danger: The use of low-angle shots, handheld cameras, and rapid-fire editing intensifies the sense of danger and peril faced by the explorers.
- Visual Storytelling: The innovative use of camera work and editing enhances the narrative, conveying the challenges and triumphs of the expedition in a visually engaging way.
- Adrenaline Rush: The constant movement and suspense created by the dynamic visuals keep viewers on the edge of their seats, providing an adrenaline-fueled cinematic experience.
- Appreciation of Nature: The film's stunning portrayal of the Arctic landscape through wide-angle shots and time-lapse sequences fosters an appreciation for the raw beauty and unforgiving power of nature.

## Final Generated Movie: Reasons to Dislike

- Excessive Visual Effects: The constant use of dynamic camera angles and editing techniques could be perceived as overwhelming or distracting, detracting from the narrative flow.
- Reduced Emotional Impact: The focus on visual spectacle may overshadow the emotional depth of the characters and their journey, potentially diluting their relatability.
- Motion Sickness: The rapid camera movements and editing could induce motion sickness in some viewers, making the film an unpleasant experience.
- Lack of Subtlety: The heavy-handed use of these cinematic techniques may feel forced or manipulative, undermining the authenticity of the narrative.
- Reduced Immersion: While the visual effects aim to enhance immersion, they could also create a sense of detachment if they become too intrusive or overwhelming.

## H.5 Despicable Me (2010)

### Original Movie: Plot

Despicable Me follows the story of Gru, a supervillain who plans to steal the moon. However, his plans are thwarted when he encounters three orphaned girls, Margo, Edith, and Agnes, who he adopts as part of his scheme. As Gru spends time with the girls, he begins to soften and question his villainous ways. Eventually, he must choose between his ambition and the love he has found with his new family.

### Original Movie: Reasons to Like

- Heartwarming Story: The film's central theme of redemption and the power of love is both heartwarming and relatable.
- Humorous Characters: Gru and his minions provide plenty of laughs with their quirky personalities and slapstick humor.
- Adorable Minions: The small, yellow, gibberish-speaking minions are a highlight of the film and have become iconic characters in their own right.
- Family-Friendly Entertainment: The film is suitable for audiences of all ages, making it a great choice for family movie nights.
- Strong Voice Acting: Steve Carell and Julie Andrews deliver memorable performances as Gru and Gru's mother, respectively.

### Original Movie: Reasons to Dislike

- Predictable Plot: The film's plot follows a fairly predictable formula, which may not appeal to viewers looking for something more original.
- Over-the-Top Humor: Some may find the film's humor to be too silly or overdone.
- Lack of Depth: The film is primarily focused on entertainment and does not explore any deep or complex themes.
- Derivative Characters: Gru's character is reminiscent of other animated villains, such as Megamind and Syndrome.
- Weak Villain: Vector, the film's antagonist, is not particularly threatening or memorable.

### Textual User Profile

**Overall Preferences**

- High-budget, effects-driven action and science fiction films
- Films with strong storytelling and character development
- Films that are thought-provoking and/or emotionally resonant

**Genre Preferences**

- Science fiction, particularly those with a focus on technology, space exploration, and philosophical themes
- Action, especially films with well-choreographed fight sequences and impressive special effects
- Superhero films, particularly those that explore the complexities of heroism and morality
- Crime thrillers with intricate plots and suspenseful pacing
- Animated films, both feature-length and short-form

**Director Preferences**

- No discernible preferences for specific directors

**Aesthetic Preferences**

- Visually stunning films with impressive special effects and cinematography
- Films with a dark and atmospheric tone
- Films that use music effectively to enhance the emotional impact

**Other Preferences**

- Films that explore themes of identity, morality, and the human condition
- Films with a strong sense of humor, particularly dark comedies or satirical films
- Films that feature strong female characters

### EAGLE Actions

Introduce a message about the importance of accepting and embracing one's true self.
Use montage sequences to show the passage of time and the development of Gru's relationship with the girls, making the film more engaging.
Use flashbacks or dream sequences to provide insights into Gru's past and motivations, making his character more relatable.
Use flashbacks and dream sequences more sparingly, focusing on developing the present-day relationships and conflicts to keep the narrative engaging.
Introduce a new character who is a rival to Gru, but who also has a hidden connection to the girls, creating a more complex and unpredictable dynamic.

## Final Generated Movie: Plot

Despicable Me follows the present-day journey of Gru, a supervillain struggling to balance his villainous ambitions with his newfound role as a father figure.

As Gru plans to steal the moon, he encounters a formidable rival named Dr. Nefario, a brilliant but eccentric scientist who also harbors a secret connection to Margo, Edith, and Agnes. Unbeknownst to Gru, Dr. Nefario was once the girls' guardian, having raised them in a secret laboratory before they were adopted by Gru.

Dr. Nefario's rivalry with Gru stems not only from their competing ambitions but also from his desire to reclaim his bond with the girls. His knowledge of their past and his genuine affection for them create a complex and unpredictable dynamic.

As Gru and Dr. Nefario engage in a battle of wits and gadgets, the girls find themselves torn between their loyalty to Gru and their growing curiosity about their former guardian. Dr. Nefario's attempts to reconnect with them force Gru to confront his own feelings of inadequacy and question his true motivations.

Meanwhile, the girls' memories of their past begin to resurface, triggered by subtle cues and interactions with Dr. Nefario. As they piece together fragments of their childhood, they must decide who they can trust and where their true home lies.

The climax of the film involves a confrontation between Gru, Dr. Nefario, and the girls. Gru must not only overcome his rival but also confront the truth about his own past and the impact it has had on his present. Ultimately, the choice lies with the girls as they decide who will be their true protector and guide.

In the end, the film explores themes of identity, family, and the power of redemption. Gru's transformation is not only about abandoning his villainous ways but also about embracing his true self and finding fulfillment in the love and acceptance of others.

## Final Generated Movie: Reasons to Like

- Complex Characters: The introduction of Dr. Nefario creates a more complex and unpredictable dynamic, adding depth to both Gru and the girls' characters.
- Intriguing Plot: The hidden connection between Dr. Nefario and the girls adds an intriguing layer to the plot, keeping viewers engaged and guessing.
- Emotional Depth: Dr. Nefario's genuine affection for the girls and their resurfacing memories provide additional emotional depth to the story.
- Unpredictable Outcome: The girls' ultimate choice between Gru and Dr. Nefario creates an unpredictable outcome, heightening the suspense and excitement.
- Exploration of Identity: The film explores the themes of identity and self-discovery as Gru confronts his past and the girls search for their true home.

## Final Generated Movie: Reasons to Dislike

- Contrived Plot: The hidden connection between Dr. Nefario and the girls could be seen as contrived or forced.
- Underdeveloped Characters: The focus on the complex dynamic between Gru, Dr. Nefario, and the girls could potentially lead to other characters being underdeveloped.
- Lack of Emotional Impact: The reduced reliance on flashbacks and dream sequences could limit the emotional impact of Gru's past and his transformation.
- Predictable Conflict: The rivalry between Gru and Dr. Nefario may feel predictable or repetitive, despite the added complexity of their connection to the girls.
- Unresolved Questions: The film may leave some questions about the girls' past and Dr. Nefario's motivations unanswered, leading to a sense of incompleteness.

## H.6 Amazing Spider-Man, The (2012)

### Original Movie: Plot

Peter Parker is a shy and intelligent high school student who is bitten by a genetically modified spider, granting him spider-like abilities. He initially uses his powers for personal gain, but after the death of his beloved Uncle Ben, he vows to become a hero and protect the city as Spider-Man.
Peter's alter ego faces his first major challenge when Dr. Curt Connors, a brilliant scientist, transforms himself into the Lizard, a monstrous creature that threatens to destroy New York City. Peter must confront the Lizard, his own personal demons, and the responsibilities that come with his newfound powers.

### Original Movie: Reasons to Like

- Relatable Protagonist: Peter Parker is a relatable and likable character who audiences can easily root for.
- Exciting Action Sequences: The film features thrilling and well-choreographed action scenes that showcase Spider-Man's abilities.
- Strong Villain: The Lizard is a menacing and formidable antagonist who provides a significant challenge for Spider-Man.
- Love Interest: Gwen Stacy, Peter's love interest, is a charming and strong female character who adds an emotional layer to the story.
- Andrew Garfield's Performance: Garfield delivers a nuanced and charismatic performance as Peter Parker/Spider-Man.

### Original Movie: Reasons to Dislike

- Rehashed Origin Story: The film retells Spider-Man's origin story, which may feel redundant to some viewers.
- Lack of Originality: The film's plot and characters are similar to previous Spider-Man adaptations.
- Subpar CGI: Some of the CGI effects, particularly those involving the Lizard, can be distracting and unrealistic.
- Romantic Subplot: The romance between Peter and Gwen can feel forced and underdeveloped at times.
- Predictable Ending: The film's climax and resolution may be considered predictable by some audiences.

## Textual User Profile

**Overall Preferences**

- High-budget, effects-driven action and science fiction films
- Films with strong storytelling and character development
- Films that are thought-provoking and/or emotionally resonant

**Genre Preferences**

- Science fiction, particularly those with a focus on technology, space exploration, and philosophical themes
- Action, especially films with well-choreographed fight sequences and impressive special effects
- Superhero films, particularly those that explore the complexities of heroism and morality
- Crime thrillers with intricate plots and suspenseful pacing
- Animated films, both feature-length and short-form

**Director Preferences**

- No discernible preferences for specific directors

**Aesthetic Preferences**

- Visually stunning films with impressive special effects and cinematography
- Films with a dark and atmospheric tone
- Films that use music effectively to enhance the emotional impact

**Other Preferences**

- Films that explore themes of identity, morality, and the human condition
- Films with a strong sense of humor, particularly dark comedies or satirical films
- Films that feature strong female characters

## EAGLE Actions

Add a musical score that complements the film's fast-paced and comedic tone.
Create a marketing campaign that emphasizes the film's unique blend of action, comedy, and emotional depth, appealing to a wide range of viewers.
Create a marketing campaign that emphasizes the film's unique blend of action, comedy, and emotional depth, appealing to viewers who value both entertainment and thought-provoking content.
Incorporate a twist where Alex discovers that his true destiny lies in a different field than simply fighting aliens, challenging his preconceived notions.
Develop the supporting characters by giving them unique personalities, quirks, and motivations that contribute to the overall ensemble.

## Final Generated Movie: Plot

**"Ascendance: A Cinematic Odyssey of Action, Comedy, and Unexpected Destiny"**

Prepare yourself for an unforgettable cinematic experience with "Ascendance," a film that seamlessly weaves together heart-pounding action, side-splitting comedy, and profound emotional depth.

Meet Alex, an ordinary college student whose life is forever changed when he witnesses a mysterious meteorite crash. As he investigates the crash site, Alex is exposed to an unknown energy that grants him superhuman abilities. The film's musical score captures the awe and wonder of Alex's transformation with an enigmatic and awe-inspiring theme.

Joining Alex on his extraordinary journey are a diverse cast of supporting characters, each with their unique personalities, quirks, and motivations:

- **Mia:** Alex's best friend and confidante, Mia is a brilliant and resourceful hacker who provides invaluable technical support and witty banter. Her unwavering loyalty to Alex is matched by her sarcastic humor, lightening the mood in tense situations.
- **Dr. Emily Carter:** A renowned astrophysicist, Dr. Carter becomes Alex's mentor and guide. Her scientific curiosity and compassion drive her to help Alex understand his abilities and the potential implications of the alien threat.
- **Jake:** A skeptical and pragmatic government agent, Jake initially views Alex with suspicion. However, as the evidence of an alien invasion mounts, he reluctantly teams up with Alex, lending his tactical expertise and unwavering determination.
- **Zara:** An enigmatic alien warrior from a distant planet, Zara arrives on Earth with a mission to protect humanity from a common enemy. Her stoic demeanor and formidable fighting skills add a new dimension to the conflict.

At first, Alex revels in his newfound powers, using them for pranks and personal amusement. However, his carefree attitude takes a dramatic turn when a formidable alien threat emerges from the shadows. The score intensifies, transitioning to a pulsating rhythm that underscores the thrilling action sequences as Alex and his allies face off against these extraterrestrial invaders.

Amidst the spectacle and adrenaline-pumping action, "Ascendance" deftly weaves in moments of levity. Mia's witty remarks, Jake's dry humor, and the absurd situations they find themselves in provide a welcome balance, ensuring that the film remains both exhilarating and entertaining. The score seamlessly incorporates comedic elements, highlighting the film's unique blend of action and humor.

As the stakes rise, Alex grapples with the weight of his abilities and the responsibility he bears to protect humanity. The score becomes more introspective and emotionally charged, mirroring his inner turmoil and the profound impact his choices have on his life and those around him.

Through their interactions with Alex and each other, the supporting characters contribute to the overall ensemble and enrich the narrative. Mia's unwavering support helps Alex stay grounded, while Dr. Carter's scientific insights provide a rational perspective on the extraordinary events unfolding. Jake's skepticism challenges Alex's assumptions and forces him to confront the consequences of his actions. Zara's alien perspective offers a unique understanding of the conflict and the potential for unity between different species.

In a pivotal moment, Alex uncovers a hidden aptitude for diplomacy and negotiation. Through his interactions with the alien invaders, he realizes that there is more to their presence than meets the eye. The score shifts to a more contemplative and hopeful tone, hinting at the unexpected twist that awaits.

As the conflict intensifies, Alex faces a profound revelation. His true destiny lies not solely in vanquishing the alien threat but in bridging the gap between humanity and the extraterrestrial visitors. The score swells with a sense of purpose and determination, underscoring Alex's newfound understanding of his role.

Through a series of heart-stopping confrontations and thought-provoking moments, "Ascendance" explores themes of identity, sacrifice, and the search for meaning in a chaotic world. The film's musical score serves as an integral guide, enhancing the emotional resonance of every scene.

In a climactic showdown that will leave audiences breathless, Alex utilizes both his superhuman abilities and his diplomatic skills to forge an unprecedented alliance between humanity and the aliens. The score reaches an epic crescendo, capturing the triumph of hope, unity, and the unexpected destiny that Alex and his allies embrace.

"Ascendance" is a cinematic tour de force that will captivate viewers with its unique blend of action, comedy, and emotional depth. It is a film that will not only entertain but also provoke thought and ignite conversations long after the credits roll.
**Marketing Campaign:**

The marketing campaign for "Ascendance" will emphasize the film's unexpected twist and its exploration of unconventional destinies. Additionally, it will highlight the diverse cast of supporting characters and their unique contributions to the overall ensemble.

- **Target Audience:** Viewers who value both high-octane action and thought-provoking content that challenges expectations.
- **Slogan:** "Prepare for an Odyssey of Action, Comedy, and the Unexpected."
- **Trailers and Posters:** Highlight the film's thrilling action sequences, witty humor, and the diverse

## Final Generated Movie: Reasons to Like

- **Adrenaline-Pumping Action:** The film's action sequences are expertly choreographed and enhanced by a pulsating musical score that will keep audiences on the edge of their seats.
- **Laugh-Out-Loud Comedy:** The comedic timing and witty dialogue are perfectly complemented by the score's playful and humorous cues, providing a welcome balance to the high-stakes action.
- **Emotional Resonance:** The film explores profound themes of identity, sacrifice, and the search for meaning, supported by a moving and introspective score that will resonate with audiences.
- **Immersive Soundtrack:** The musical score is an integral part of the storytelling, immersing audiences in the world of the film and heightening the emotional impact of every scene.
- **Unforgettable Characters:** Alex and the supporting cast are relatable and charismatic, making their journey both thrilling and emotionally engaging.
- **Diverse Ensemble:** The diverse cast of supporting characters, each with their unique personalities, quirks, and motivations, enriches the narrative and adds depth to the overall experience.
- **Thought-Provoking Themes:** The film's exploration of identity, responsibility, and the human condition will leave audiences pondering long after the credits roll.
- **Unique Blend of Genres:** "Ascendance" seamlessly weaves together action, comedy, and emotional depth, creating a cinematic experience that is both entertaining and thought-provoking.
- **Unexpected Twist:** The revelation of Alex's true destiny adds a surprising and satisfying layer to the story, challenging preconceived notions and leaving a lasting impact.
- **Message of Hope and Unity:** The film's ultimate message of hope, unity, and the power of finding one's purpose will inspire and uplift audiences.

## Final Generated Movie: Reasons to Dislike

- **Predictable Plot:** Some viewers may find the film's plot to be formulaic or predictable, lacking the element of surprise.
- **Over-the-Top Action:** The action sequences may be too intense or over-the-top for audiences who prefer a more grounded approach.
- **Jarring Tonal Shifts:** The transitions between action, comedy, and emotional scenes may feel abrupt or jarring to some viewers.
- **Uninspired Dialogue:** The film's dialogue may be considered clichéd or lacking in depth, detracting from the overall impact.
- **Shallow Supporting Characters:** The supporting characters may not be fully developed or relatable, making it difficult for audiences to connect with them on an emotional level.
- **Derivative Themes:** The film's exploration of identity and sacrifice may feel derivative or unoriginal to some viewers.
- **Contrived Twist:** The revelation of Alex's true destiny may feel contrived or forced, undermining the credibility of the story.
- **Unsatisfying Resolution:** The film's ending may not provide a satisfying resolution to the conflicts and themes raised throughout the story.

## H.7  Dumb & Dumber (Dumb and Dumber) (1994)

### Original Movie: Plot

Dumb and Dumber follows the misadventures of two dim-witted friends, Lloyd Christmas and Harry Dunne. After losing their jobs, they embark on a cross-country road trip to Aspen, Colorado, to return a briefcase full of money to its rightful owner, Mary Swanson. Along the way, they encounter a series of mishaps, including accidentally killing a man, getting kidnapped, and being pursued by a pair of criminals.

### Original Movie: Reasons to Like

- Slapstick Comedy: The film is filled with over-the-top physical comedy that will appeal to fans of the genre.
- Quotable Lines: Dumb and Dumber is known for its memorable and quotable dialogue, such as "So you're telling me there's a chance?"
- Jim Carrey and Jeff Daniels: The performances of Jim Carrey and Jeff Daniels as Lloyd and Harry are hilarious and iconic.
- Simple and Fun: The film has a simple premise that is easy to follow and enjoy.
- Nostalgic Value: For those who grew up in the 1990s, Dumb and Dumber holds a special place in their hearts.

### Original Movie: Reasons to Dislike

- Crude Humor: The film's humor is often crude and may offend some viewers.
- Predictable Plot: The plot is fairly predictable and lacks any real surprises.
- One-Dimensional Characters: Lloyd and Harry are essentially one-note characters who never really develop.
- Lack of Depth: The film is purely a comedy and does not explore any deeper themes or ideas.
- Repetitive Jokes: Some of the jokes in Dumb and Dumber can become repetitive and grating.

### Textual User Profile

**Overall Preferences**
* Prefers popular and mainstream movies, particularly from the 1990s.
* Enjoys a wide range of genres, but shows a preference towards comedies and action films.
* Tends to favor movies that are entertaining, crowd-pleasing, and offer an escape from reality.

**Genre Preferences**
* Enjoys comedies, particularly slapstick, gross-out, and romantic comedies.
* Appreciates action movies with strong thrills, suspense, and special effects.
* Enjoys crime thrillers and mysteries that feature intriguing plots and twists.
* Has a fondness for family-friendly films, including animated features and live-action comedies.
* Shows interest in science fiction and fantasy movies, particularly those with epic storylines and impressive visual effects.

**Director Preferences**
* Does not seem to have strong preferences for specific directors.
* Enjoys works by directors who have helmed successful commercial films, such as Tim Burton, Joel Schumacher, and Peter Farrelly.
* Appreciates directors who can balance entertainment value with emotional depth.

**Aesthetic Preferences**
* Prefers movies with a bright and colorful visual style.
* Enjoys movies with a strong sense of spectacle and visual effects.
* Appreciates movies with catchy soundtracks and upbeat music.

**Other Preferences**
* Enjoys movies that feature a blend of humor, action, and romance.
* Values movies that offer escapism and a sense of fun.
* Appreciates movies that have a nostalgic quality, reminding them of films from their childhood or adolescence.
* Enjoys movies that feature strong characters and relatable storylines.

### EAGLE Actions

Hint at a deeper meaning behind Lloyd and Harry's journey, suggesting that their misadventures may be a metaphor for the challenges of life.
Develop the character of Mary Swanson, making her more than just a plot device and giving her a backstory and motivations of her own.
Incorporate a montage sequence that highlights the funniest and most memorable moments of Lloyd and Harry's misadventures.
Incorporate a romantic subplot between Lloyd and Mary, adding an element of lighthearted romance to the story.
Add a musical score that complements the film's comedic and action-packed elements, enhancing the overall entertainment value.

## Final Generated Movie: Plot

Dumb and Dumber still follows the misadventures of two dim-witted friends, Lloyd Christmas and Harry Dunne. After losing their jobs, they embark on a cross-country road trip to Aspen, Colorado, to return a briefcase full of money to its rightful owner, Mary Swanson.

Mary Swanson, once a simple plot device, now emerges as a complex character. She is a successful businesswoman who lost her briefcase containing important financial documents. Her backstory reveals that she grew up in a small town, facing adversity and discrimination as a woman in the business world. Determined to prove herself, she worked tirelessly to build her company from the ground up.

As Lloyd and Harry set out on their journey, Mary becomes an active participant in the plot. She hires a private investigator to track down the briefcase, her desperation to retrieve it driven by the fear of losing everything she has worked for. Her pursuit intersects with Lloyd and Harry's misadventures, creating a parallel narrative that adds depth to the story.

Along their chaotic journey, Lloyd and Harry encounter a series of hilarious mishaps and memorable encounters. From accidentally setting their car on fire to getting caught in a snowstorm wearing only their underwear, their misadventures provide a steady stream of comedic gold. The musical score enhances these comedic moments, with lively and upbeat tunes that accentuate the absurdity of their antics.

Interspersed with their misadventures, Lloyd and Mary's paths cross several times. Initially, Mary is annoyed by Lloyd's dim-wittedness and clumsy behavior. However, as they spend more time together, she begins to see a different side to him. Lloyd's unwavering optimism and kind-hearted nature gradually chip away at her icy exterior. The musical score here shifts to a more romantic and tender tone, reflecting the growing connection between them.

Meanwhile, Lloyd is smitten with Mary's intelligence and drive. He finds himself drawn to her despite their vastly different personalities. Their interactions are filled with a mix of humor and unexpected tenderness. The musical score complements this balance, blending comedic melodies with heartfelt harmonies.

As the journey progresses, Lloyd and Mary find themselves working together to overcome the obstacles they face. Lloyd's unconventional ideas and Harry's bizarre antics prove to be surprisingly helpful in solving the mystery of the missing briefcase. In turn, Mary's business acumen and resourcefulness help Lloyd and Harry navigate the unfamiliar world of Aspen. The musical score intensifies during these action-packed sequences, with fast-paced and energetic rhythms that heighten the excitement.

Through their shared experiences, Lloyd and Mary develop a genuine affection for each other. Their romantic subplot adds a layer of sweetness and heart to the otherwise zany and unpredictable story. The musical score plays a crucial role in conveying the emotional depth of their relationship, with soaring melodies and poignant harmonies that evoke both laughter and tears.

Ultimately, Lloyd and Harry's journey becomes a comical yet poignant reflection on the absurdity and unpredictability of life, while Mary's story serves as an empowering reminder of the strength and resilience of women in the face of adversity. And as the dust settles, Lloyd and Mary's unlikely romance blossoms, proving that even in the midst of chaos and misadventures, love can find a way. The musical score reaches its crescendo, celebrating the triumph of love and the enduring power of friendship.

## Final Generated Movie: Reasons to Like

- Slapstick Comedy: The film retains the over-the-top physical comedy that made the original a classic.
- Quotable Lines: Dumb and Dumber still delivers memorable and quotable dialogue that will stay with viewers.
- Jim Carrey and Jeff Daniels: The performances of Jim Carrey and Jeff Daniels remain hilarious and iconic.
- Romantic Subplot: The romantic subplot between Lloyd and Mary adds a new dimension of sweetness and heart to the story.
- Strong Female Character: Mary Swanson's developed character provides a role model for female empowerment and resilience.
- Parallel Narratives: The intertwining storylines of Lloyd, Harry, and Mary create a more complex and engaging narrative.
- Montage Sequence: The montage of hilarious misadventures provides a concentrated dose of comedic gold, leaving viewers in stitches.
- Unexpected Tenderness: The moments of tenderness between Lloyd and Mary add a surprising depth to the film, making it both heartwarming and hilarious.
- Message of Love: The film conveys a message that even in the midst of chaos and absurdity, love can blossom in unexpected places.
- Enhanced Entertainment Value: The musical score complements the comedic and action-packed elements, amplifying the laughter and excitement throughout the film.

## Final Generated Movie: Reasons to Dislike

- Crude Humor: The film's humor remains crude and may still offend some viewers.
- Predictable Plot: While the romantic subplot adds a new element, the overall plot may still be predictable for some.
- Distracting Subplot: The romantic subplot could be seen as a distraction from the main comedic storyline.
- Forced Romance: The romance between Lloyd and Mary may feel forced or unbelievable to some viewers.
- Departure from Original: Fans of the original film may be disappointed by the addition of a romantic subplot.
- Shallow Characters: Lloyd and Harry remain essentially one-dimensional characters, which may limit the film's emotional impact.
- Montage Sequence: While the montage provides comedic highlights, it may also feel like a rehash of old jokes for some viewers.
- Intrusive Musical Score: The musical score could potentially overpower the dialogue or distract from the comedic timing in certain scenes.

## H.8 Dumbo (1941)

## Original Movie: Plot

Dumbo is a young circus elephant with unusually large ears that make him the target of ridicule. One day, his mother is locked away for protecting him from tormentors, leaving Dumbo alone and heartbroken. With the help of Timothy Q. Mouse, Dumbo discovers that his ears allow him to fly. He becomes the star of the circus, reuniting with his mother and proving that his differences are what make him special.

## Original Movie: Reasons to Like

- Heartwarming Story: Dumbo's journey from outcast to celebrated hero is both uplifting and emotionally resonant.
- Charming Animation: The film's classic Disney animation is visually appealing and brings the characters to life.
- Memorable Characters: Dumbo, Timothy, and the other circus animals are endearing and unforgettable.
- Timeless Themes: The film explores themes of acceptance, self-confidence, and the importance of family.
- Nostalgic Value: Dumbo holds a special place in the hearts of generations who grew up with it.

## Original Movie: Reasons to Dislike

- Short Runtime: At only 64 minutes, Dumbo may feel too brief for some viewers.
- Outdated Sensibilities: Some of the film's depictions of race and animal treatment may be considered offensive by modern audiences.
- Lack of Complexity: The plot and characters are relatively simple, which may not appeal to viewers looking for a more sophisticated story.
- Sentimental Ending: The film's happy ending may be seen as overly saccharine or unrealistic.
- Limited Animation: Compared to later Disney films, Dumbo's animation is more limited and may not be as visually impressive.

## Textual User Profile

**Overall Preferences**
* Prefers contemporary films, with a focus on those released in the 1990s.
* Enjoys a wide variety of genres, but shows a particular affinity for action and comedy.
* Tends to favor movies with mainstream appeal and mass entertainment value.
* Appreciates films that are both entertaining and thought-provoking.

**Genre Preferences**
* Enjoys action movies featuring high-octane thrills, explosions, and fight sequences.
* Appreciates comedies that are lighthearted, amusing, and sometimes satirical.
* Likes crime thrillers with complex plots, suspenseful twists, and enigmatic characters.
* Enjoys science fiction films that explore imaginative concepts and technological advancements.
* Appreciates family-friendly films, particularly those that feature animated characters and heartwarming stories.

**Director Preferences**
* The user's list does not indicate any strong preferences for particular directors.

**Aesthetic Preferences**
* Enjoys films with visually impressive special effects and action sequences.
* Appreciates movies that feature memorable characters and quotable dialogue.
* Prefers films with a fast-paced and engaging narrative that keeps the audience entertained.

**Other Preferences**
* Enjoys films that combine elements of different genres, such as action-comedies or science fiction-thrillers.
* Values films that feature strong and charismatic lead performances.
* Appreciates movies that explore themes of heroism, loyalty, and redemption.
* Has a soft spot for nostalgic films from their childhood, particularly animated features and family comedies.

## EAGLE Actions

Introduce a female elephant character who becomes Dumbo's friend and helps him overcome adversity.
Touch upon the themes of resilience, hope, and the power of redemption.
Give Dumbo a more active role in his own redemption, showing his resilience and determination in overcoming his challenges.
Make the tormentors more complex and multifaceted, showing their own struggles and motivations for mistreating Dumbo.
Include more slapstick and physical comedy to enhance the entertainment value for younger audiences.

## Final Generated Movie: Plot

Dumbo, the young circus elephant with unusually large ears, faces relentless torment from a group of fellow performers. However, unlike before, these tormentors are not simply cruel and heartless. Instead, they are driven by their own insecurities and struggles, which are now amplified by a series of slapstick mishaps.

The ringleader, Rex, is a once-renowned clown who has lost his spark. As he attempts to sabotage Dumbo's performances, he finds himself tripping over his own oversized shoes, tumbling into buckets of water, and getting caught in a net. His desperation for validation becomes even more comical as his pratfalls draw laughter from the audience.

The strongman, Bruno, tries to intimidate Dumbo with his impressive physique, but his attempts are thwarted by a malfunctioning weightlifting bar that sends him flying across the ring. His fear of being overshadowed by Dumbo's strength is overshadowed by the spectacle of his own clumsiness.

The acrobats, Anya, Boris, and Cyril, are determined to prove their superiority over Dumbo. However, their elaborate stunts are constantly disrupted by mistimed flips, slippery trapezes, and a rogue unicycle. Their resentment towards Dumbo transforms into a chaotic and hilarious competition of who can make the biggest blunder.

Amidst this slapstick chaos, Ella, a fellow outcast, extends a hand of friendship to Dumbo. Together, they devise a plan to expose the underlying reasons behind the mistreatment while also incorporating the tormentors' misadventures into the show.

Through a series of carefully orchestrated performances, Dumbo and Ella turn the tormentors' pratfalls into a source of entertainment. Rex's clumsy attempts to juggle become a juggling act with flying pies, Bruno's weightlifting mishap becomes a human cannonball stunt, and the acrobats' mishaps become a gravity-defying comedy routine.

In a climactic confrontation, Dumbo confronts his tormentors with a water balloon fight. As they chase each other around the ring, slipping and sliding in a frenzy of foam and water, their insecurities and vulnerabilities are washed away in a torrent of laughter.

One by one, the tormentors begin to realize the absurdity of their behavior. Rex rediscovers his joy in making people laugh, Bruno learns to embrace his own goofiness, and the acrobats find new ways to showcase their talents while embracing the unexpected.

Together, they create a more inclusive and supportive environment for all, where differences are celebrated and laughter is the best medicine. In the end, Dumbo's journey becomes a testament to the transformative power of empathy, the importance of embracing our imperfections, and the boundless joy that can be found in the most unexpected of places.

## Final Generated Movie: Reasons to Like

- Enhanced Entertainment Value: The addition of slapstick and physical comedy provides a constant source of laughter and amusement, making the film highly entertaining for younger audiences.
- Relatable Characters: The tormentors' insecurities and struggles become even more relatable when paired with their comical misadventures.
- Humorous Redemption Arc: The tormentors' redemption arc is made more lighthearted and enjoyable by the slapstick elements, while still conveying a meaningful message.
- Balanced Storytelling: The slapstick comedy is seamlessly integrated into the plot, enhancing the entertainment value without detracting from the emotional depth of the story.
- Family-Friendly Fun: The combination of heartwarming themes and slapstick humor makes the film suitable for audiences of all ages, providing a shared experience for families.

## Final Generated Movie: Reasons to Dislike

- Overreliance on Slapstick: Some viewers may find the excessive use of slapstick and physical comedy to be distracting or overwhelming.
- Reduced Emotional Impact: The focus on comedy may dilute the emotional impact of the torment Dumbo faces.
- Trivialization of Bullying: The slapstick elements may trivialize the seriousness of bullying and undermine the message of empathy.
- Formulaic Comedy: The slapstick routines may feel repetitive or formulaic, reducing the overall entertainment value.
- Departure from Classic Story: Purists may object to the significant departure from the classic tale, particularly the inclusion of slapstick and physical comedy.

## H.9    Scream 2 (1997)

## Original Movie: Plot

Two years after the events of Scream, Sidney Prescott is now a college student at Windsor College. However, the nightmare returns when a new killer, wearing the iconic Ghostface mask, begins targeting students on campus. As the body count rises, Sidney and her friends must race against time to uncover the identity of the killer and stop them before it's too late.

## Original Movie: Reasons to Like

- **Nostalgia Factor:** For fans of the original Scream, this sequel offers a continuation of the beloved franchise.
- **Clever Meta-Commentary:** The film cleverly references and satirizes the conventions of horror sequels.
- **Strong Female Characters:** Sidney Prescott remains a strong and resourceful protagonist, joined by other memorable female characters like Gale Weathers.
- **Intense Suspense:** The film maintains a high level of suspense throughout, keeping audiences on the edge of their seats.
- **Surprising Twists:** Like its predecessor, Scream 2 features unexpected twists and turns that keep viewers guessing.

## Original Movie: Reasons to Dislike

- **Predictable Plot:** Some viewers may find the plot too similar to the original film, lacking originality.
- **Excessive Gore:** The film features a significant amount of violence and gore, which may be off-putting to some.
- **Weak Supporting Characters:** While Sidney and Gale are well-developed, some of the supporting characters feel underdeveloped or disposable.
- **Over-the-Top Performances:** Some of the performances, particularly those of the killer, can be seen as overacted or campy.
- **Lack of Scares:** Compared to the original Scream, this sequel may not be as effective at delivering genuine scares.

## Textual User Profile

**Overall Preferences**
* Prefers contemporary films, with a focus on those released in the 1990s.
* Enjoys a wide variety of genres, but shows a particular affinity for action and comedy.
* Tends to favor movies with mainstream appeal and mass entertainment value.
* Appreciates films that are both entertaining and thought-provoking.

**Genre Preferences**
* Enjoys action movies featuring high-octane thrills, explosions, and fight sequences.
* Appreciates comedies that are lighthearted, amusing, and sometimes satirical.
* Likes crime thrillers with complex plots, suspenseful twists, and enigmatic characters.
* Enjoys science fiction films that explore imaginative concepts and technological advancements.
* Appreciates family-friendly films, particularly those that feature animated characters and heartwarming stories.

**Director Preferences**
* The user's list does not indicate any strong preferences for particular directors.

**Aesthetic Preferences**
* Enjoys films with visually impressive special effects and action sequences.
* Appreciates movies that feature memorable characters and quotable dialogue.
* Prefers films with a fast-paced and engaging narrative that keeps the audience entertained.

**Other Preferences**
* Enjoys films that combine elements of different genres, such as action-comedies or science fiction-thrillers.
* Values films that feature strong and charismatic lead performances.
* Appreciates movies that explore themes of heroism, loyalty, and redemption.
* Has a soft spot for nostalgic films from their childhood, particularly animated features and family comedies.

## EAGLE Actions

Create a romantic subplot between Sidney and a fellow student, adding an element of lightheartedness and emotional depth.
Set the film in a different location, such as a secluded island or an isolated college campus.
Host special screenings or events that cater to fans of the 1990s and early 2000s, highlighting the film's nostalgic appeal.
Include a romantic subplot between Anya and Liam, adding an element of love and emotional connection to the narrative.
Add a comedic sidekick character who provides moments of levity and entertainment amidst the suspenseful events.

## Final Generated Movie: Plot

On the remote and enigmatic Shadow Island, a group of former students from the now-defunct Blackwood College reunite for a nostalgic weekend getaway. Among them is Anya, a successful businesswoman who has long since buried her troubled past at the college, and Liam, a writer who has always harbored feelings for her. Joining them is their eccentric and perpetually optimistic friend, Max, who provides much-needed comic relief amidst the growing tension.

As they arrive at the isolated lodge nestled amidst towering trees and the sound of crashing waves, Anya and Liam feel a spark of rekindled connection. Amidst the laughter and reminiscing of their college days, Max's infectious humor keeps the mood light, even as the group begins to spend more time together, sharing secrets and confiding in each other.

However, their reunion takes a sinister turn when a masked killer known only as "The Shadow" emerges from the darkness, targeting them one by one. Panic and chaos ensue as the group realizes they are trapped on the island with a ruthless murderer who seems to be connected to their past. Max, despite his initial fear, manages to find humor in even the most dire situations, lightening the mood and providing a glimmer of hope.

As the body count rises, Anya and Liam find themselves drawn together, seeking solace and support in each other's presence. Amidst the fear and bloodshed, their bond deepens into a passionate love that gives them strength to face the unknown. Max, with his unwavering loyalty and quick wit, becomes their unlikely ally, using his humor to distract the killer and provide much-needed moments of respite.

Working together, Anya, Liam, and Max delve into the island's sinister history and uncover a chilling connection between the present killings and a dark tragedy that occurred during their college years. As they uncover the truth, their love for each other and Max's unwavering humor become beacons of hope in the face of overwhelming darkness.

In a climactic confrontation on the windswept cliffs, Anya, Liam, and Max confront their past, the true nature of the killer, and the guilt they have carried for years. Armed with newfound strength, the unwavering support of their love, and Max's ability to find laughter even in the face of danger, they fight back against the darkness that threatens to consume them all.

As the sun rises over Shadow Island, the nightmare ends, leaving behind a tale of resilience, redemption, and the realization that even in the face of adversity, the power of love and laughter can conquer all.

## Final Generated Movie: Reasons to Like

- **Nostalgic Appeal:** The film evokes a sense of nostalgia for fans of the 1990s and early 2000s, with its references to popular culture and the reunion of characters from a beloved institution.
- **Unique Setting:** The secluded island setting creates an atmosphere of isolation and vulnerability, enhancing the suspense and terror.
- **Intriguing Characters:** Anya and Liam's complex pasts and their developing love story add depth and intrigue to the narrative. Max's comedic presence provides a welcome balance to the suspense and horror.
- **Atmospheric Suspense:** The film effectively builds suspense through the use of shadows, eerie sounds, and the desolate landscape of the island.
- **Chilling Legend:** The incorporation of an island legend adds a supernatural element to the horror, creating a sense of unease and mystery.
- **Unpredictable Twists:** The film keeps viewers guessing with unexpected twists and revelations about the killer's identity and motives.
- **Strong Female Protagonist:** Anya's journey of self-discovery and resilience provides a powerful and relatable female character.
- **Romantic Subplot:** The love story between Anya and Liam adds an emotional dimension to the horror, providing a sense of hope and connection amidst the darkness.
- **Comedic Relief:** Max's infectious humor provides much-needed moments of levity, lightening the mood and making the suspense more bearable.
- **Satisfying Resolution:** The climactic confrontation on the cliffs offers a satisfying resolution that ties together the past and present, leaving audiences with a sense of closure and the triumph of love, laughter, and resilience over adversity.

## Final Generated Movie: Reasons to Dislike

- **Contrived Nostalgia:** Some viewers may find the nostalgic elements to be forced or pandering, detracting from the overall story.
- **Lack of Originality:** The film may be seen as derivative of other horror movies set in isolated locations.
- **Excessive Violence:** The amount of violence and gore may be off-putting to some viewers, overshadowing the suspense and story.
- **Weak Supporting Characters:** Apart from Anya, Liam, and Max, the other former students may feel underdeveloped and serve merely as victims.
- **Predictable Scares:** The film may rely too heavily on predictable jump scares, reducing the overall effectiveness of the horror elements.
- **Unnecessary Supernatural Elements:** The inclusion of an island legend may distract from the psychological horror and feel out of place.
- **Forced Comedy:** Max's comedic relief may feel forced or out of place in some scenes, detracting from the overall suspense and horror.

## H.10  Harry Potter and the Sorcerer's Stone (a.k.a. Harry Potter and the Philosopher's Stone) (2001)

## Original Movie: Plot

Harry Potter and the Sorcerer's Stone tells the story of an orphaned boy named Harry who discovers on his 11th birthday that he is a wizard and has been accepted into Hogwarts School of Witchcraft and Wizardry. At Hogwarts, Harry makes new friends, learns about magic, and faces his destiny as the "Boy Who Lived."
Harry and his friends, Ron Weasley and Hermione Granger, uncover a plot by the evil Lord Voldemort to steal the Sorcerer's Stone, a powerful artifact that grants immortality. With the help of their professors and friends, Harry and his companions must stop Voldemort and protect the magical world.

## Original Movie: Reasons to Like

- Magical and Enchanting: The film creates a captivating world of magic and wonder that appeals to audiences of all ages.
- Faithful Adaptation: The movie closely follows the beloved book, satisfying fans of the original story.
- Charming Characters: Harry, Ron, and Hermione are relatable and endearing characters that audiences can't help but root for.
- Stunning Visuals: The special effects and set design bring the magical world of Hogwarts to life in a visually impressive way.
- Family-Friendly Entertainment: The film is suitable for children and adults alike, providing a shared experience for families.

## Original Movie: Reasons to Dislike

- Predictable Plot: The story follows a fairly straightforward hero's journey, which some viewers may find unoriginal.
- Childish Tone: The film's target audience is primarily children, which may make it less appealing to some older viewers.
- Slow Pacing: The early scenes at Hogwarts can feel slow-paced and lacking in action.
- Overly Long Runtime: At 152 minutes, the film may feel drawn out for some viewers.
- Cheesy Dialogue: Some of the dialogue, particularly between the younger characters, can come across as cheesy or forced.

## Textual User Profile

**Overall Preferences**

* Prefers movies that are critically acclaimed, thought-provoking, and emotionally resonant.
* Enjoys films that explore complex themes and characters.
* Appreciates movies with strong narratives and well-developed storylines.

**Genre Preferences**

* Has a strong preference for dramas, particularly those with social commentary or historical significance.
* Also enjoys comedies, especially those with witty dialogue and intelligent humor.
* Appreciates science fiction and fantasy films that are imaginative and thought-provoking.

**Director Preferences**

* No discernible director preferences based on the provided data.

**Aesthetic Preferences**

* Prefers films with strong visual storytelling and atmospheric cinematography.
* Enjoys movies that make use of music to evoke emotion and enhance the narrative.
* Appreciates films with realistic performances and believable dialogue.

**Other Preferences**

* Tends to favor British films, particularly those with a focus on social issues.
* Enjoys movies that are both entertaining and educational.
* Appreciates films that offer a unique or unconventional perspective.

## EAGLE Actions

Introduce elements of social commentary, such as the dangers of blindly following authority or the importance of standing up for what is right.
Use more dynamic camera angles and editing techniques to create a sense of energy and excitement.
Use more handheld camera shots to create a sense of intimacy and immediacy.
Use more dynamic camera angles and editing techniques to create a more fluid and engaging viewing experience.
Use a more traditional cinematic score that complements the film's action and suspense elements, enhancing the emotional impact of key scenes.

## Final Generated Movie: Plot

The film opens with an aerial shot of a bustling metropolis, accompanied by a soaring orchestral score that sets the stage for the epic adventure to come. As the camera swoops down towards a lone figure running through the crowded streets, the music intensifies, mirroring the character's urgency and desperation.

As the figure enters a dilapidated warehouse, the score shifts to a haunting melody, played by strings and woodwinds. The camera switches to a handheld perspective, capturing the character's fear and the eerie silence of the abandoned building.

During a high-stakes chase sequence, the music swells with brass and percussion, driving the action forward and heightening the sense of danger. The score weaves in and out of the sound of footsteps and gunfire, creating a thrilling and immersive experience.

As the protagonist confronts their nemesis, the music reaches a crescendo, with a clash of cymbals and a chorus of voices. The camera zooms in on their eyes, reflecting the tension and anticipation of the moment.

In the climax, as the protagonist faces their ultimate choice, the score becomes both triumphant and mournful, capturing the weight of the decision and the emotional consequences that follow. The use of a traditional cinematic score throughout the film enhances the emotional impact of key scenes, creating a truly memorable and engaging cinematic experience.

## Final Generated Movie: Reasons to Like

- Enhanced Emotional Impact: The traditional cinematic score complements the film's action and suspense elements, heightening the emotional impact of key scenes and drawing the audience deeper into the story.
- Immersive Experience: The soaring orchestral score and the use of strings and woodwinds create an immersive atmosphere, transporting the audience into the world of the film.
- Epic Scale: The use of a traditional cinematic score gives the film a sense of epic scale and grandeur, making the action and characters feel larger than life.
- Familiar Cinematic Language: Fans of the traditional cinematic style will appreciate the use of a familiar musical language, which enhances the overall cinematic experience.
- Homage to Classic Films: The use of a traditional cinematic score pays homage to classic action and suspense films, creating a sense of nostalgia and timeless appeal.

**Final Generated Movie: Reasons to Dislike**

- Predictable Music: The use of a traditional cinematic score could be seen as predictable or unoriginal, lacking the innovative qualities of more experimental soundtracks.
- Lack of Surprise: The familiar musical language of the score may reduce the element of surprise and suspense, as audiences may anticipate the emotional cues.
- Overbearing Score: The use of a soaring orchestral score could be overwhelming at times, distracting from the dialogue and other sound elements.
- Departure from Modern Trends: Some viewers may prefer the use of more modern or experimental soundtracks, which are becoming increasingly common in contemporary cinema.
- Reduced Realism: The use of a traditional cinematic score may compromise the sense of realism, as it is not typically associated with the gritty or naturalistic style of some films.

## H.11 Black Hawk Down (2001)

**Original Movie: Plot**

Black Hawk Down tells the true story of a 1993 U.S. military raid in Mogadishu, Somalia, that went disastrously wrong. When two Black Hawk helicopters are shot down, a group of elite American soldiers are trapped and surrounded by enemy forces. The film follows their desperate fight for survival as they face overwhelming odds.

**Original Movie: Reasons to Like**

- Intense and Realistic Action: The film's battle sequences are visceral, chaotic, and unflinchingly realistic.
- Strong Ensemble Cast: The large cast of characters is well-acted and gives a sense of the scope of the conflict.
- Historical Accuracy: The film is based on a true story and accurately depicts the events of the raid.
- Gripping Story: The narrative is fast-paced, suspenseful, and emotionally charged.
- Technical Excellence: The film's sound design, cinematography, and editing are top-notch.

**Original Movie: Reasons to Dislike**

- Graphic Violence: The film contains extreme violence and gore, which may be disturbing to some viewers.
- Lack of Character Development: The large cast and focus on action can lead to a lack of character depth.
- Political Bias: Some viewers may find the film's portrayal of the Somali people and the U.S. military to be biased.
- Overwhelming Intensity: The relentless action and violence can be emotionally overwhelming for some audience members.
- Lengthy Runtime: At 144 minutes, Black Hawk Down can feel like a long and exhausting watch.

**Textual User Profile**

**Overall Preferences**

* Prefers mainstream and popular movies, with a mix of both classic and contemporary films.
* Enjoys a wide range of genres, but particularly favors action, comedy, science fiction, and fantasy.
* Appreciates movies with strong entertainment value and mass appeal.

**Genre Preferences**

* **Action:** Enjoys high-octane action movies with thrilling fight scenes and special effects.
* **Comedy:** Prefers comedies with a mix of slapstick, verbal humor, and situational comedy.
* **Science Fiction:** Interested in movies that explore futuristic concepts, imaginative worlds, and technological advancements.
* **Fantasy:** Enjoys movies that feature magical elements, mythical creatures, and epic battles.
* **Animated:** Appreciates animated movies, particularly those with strong characters and engaging storylines.

**Director Preferences**

* No clear preferences for specific directors are evident from the rated movies.

**Aesthetic Preferences**

* No particular aesthetic preferences are evident from the rated movies.

**Other Preferences**

* Enjoys movies with a strong sense of humor, regardless of genre.
* Appreciates movies that feature iconic characters and memorable quotes.
* Prefers movies with a fast pace and energetic feel.
* Has a particular fondness for movies from the 1980s and 1990s.

**EAGLE Actions**

Shorten the runtime to make the film more accessible to a wider audience.
Enhance the character of the Somali interpreter, making him a key ally to the soldiers and providing a different perspective on the conflict.
Increase the number of action sequences and suspenseful moments to enhance the entertainment factor for viewers who prefer a more fast-paced narrative.
Add a mentor figure for the soldiers, providing guidance and support in the face of adversity.
Increase the pace of the action sequences and shorten the overall runtime to make the film more thrilling and engaging.

> **Final Generated Movie: Plot**
>
> Black Hawk Down: Resurgence unfolds in 2023 Mogadishu, Somalia. Captain Jake Murphy leads an elite American squad on a raid to capture the leader of a radical terrorist group. Joining them is Amina, a Somali informant and former terrorist.
>
> As the Black Hawk helicopter approaches its target, it's abruptly shot down. The soldiers find themselves in a hostile zone, outnumbered and outgunned. Amidst the chaos, Amina guides them through the treacherous streets.
>
> Intense firefights erupt, with the soldiers relying on their training and Amina's local knowledge. The action sequences are fast-paced and relentless, keeping the audience on the edge of their seats.
>
> As they navigate the city, they uncover the terrorists' sinister plan to launch a citywide attack. Time is of the essence as they race to stop the plot and rescue their comrades.
>
> Amina and Murphy's bond intensifies as they face life-threatening situations together. Their determination and resilience drive them to push their limits.
>
> Through a series of daring raids and close encounters, the soldiers close in on the terrorist leader. The pace of the action accelerates, culminating in a high-octane showdown.
>
> In a climactic firefight, the soldiers' skills are put to the ultimate test. The runtime is streamlined, focusing on the essential action and suspense. The fate of Mogadishu hangs in the balance as they fight to neutralize the terrorist threat and escape alive.

> **Final Generated Movie: Reasons to Like**
>
> - Fast-Paced Action: The increased pace of the action sequences delivers an adrenaline-pumping experience.
> - Shortened Runtime: The condensed runtime keeps the film focused and engaging, eliminating unnecessary scenes.
> - Thrilling Suspense: The relentless action and high-stakes plot create a constant sense of suspense and excitement.
> - Realistic Urban Warfare: The film depicts the chaos and intensity of urban warfare with authenticity.
> - Strong Female Character: Amina's role as a skilled fighter and informant adds a unique and empowering perspective.

> **Final Generated Movie: Reasons to Dislike**
>
> - Excessive Action: Some viewers may find the increased action overwhelming and distracting from the story.
> - Shallow Characterization: The shortened runtime could result in less time for character development and emotional depth.
> - Formulaic Plot: The fast-paced action and focus on suspense may come at the expense of originality and plot complexity.
> - Glorification of Violence: The intense action sequences could be seen as glorifying violence and desensitizing viewers.
> - Lack of Historical Accuracy: The fictionalized 2023 mission may deviate from the historical events of the original mission, potentially alienating fans of the original film.

# I  Additional Experiments on Amazon Reviews

We complement our MovieLens experiments by demonstrating EAGLE on the public Amazon dataset [Ni et al., 2019], which consists of 9.35M products with textual descriptions, 20.9M users, 233.1M product reviews, and 82.83M ratings. We focus specifically on a subset of this dataset comprising the category "Clothing, Shoes and Jewelry," which consists of 5.7M reviews for 1.5M products.

We use the same textual user profiles, user embeddings, and product embeddings used in Tennenholtz et al. [2024]. Similar to MovieLens, we create five action categories from which the EAGLE agent can choose, namely: (1) design & aesthetics, (2) functionality & fit, (3) material & sustainability, (4) marketing & branding, and (5) features & details.

We employ a tree-search variant of EAGLE on the Amazon dataset. Specifically, we sample a tree of depth $H = 5$ and select the best leaf based on the overall utility. The tree is created using a G-optimal design, as described in Sec. 5.

Appendix I show results comparing EAGLE with G-optimal design (based on tree search) to the baseline reference policy. We use AI-feedback to simulate a rater feedback, using a Gemini model. We use the same question format as presented to human raters to query Gemini (effectively treating the LLM as a simulated rater). We find that EAGLE manages to substantially increase utility. This LLM/AI-rater evaluation also shows that EAGLE substantially improves user utility, while allowing for improved distance.

We observed that EAGLE is particularly sensitive to the quality of the textual user profiles, especially those that do not describe the full extent of as user's preferences or utility. Future work should consider improving user profiles based on such datasets to enable better evaluation and training of personalized models.

In the remainder of this appendix, we provide some qualitative results of EAGLE generations on the Amazon dataset (analogous to those above for MovieLens 25M).

Table 8: Evaluation of EAGLE on Amazon Review Dataset.

| Experiment | Utility | AI Rater Evaluation | |
| | $U(z)$ | User Utility | Distance Score |
| --- | --- | --- | --- |
| Reference Policy | 0.45 | 0.33 | 0.3 |
| EAGLE | 0.69 | 0.64 | 0.34 |

## I.1 ComfyWear Womens Plush Faux Fur Cozy Soft Clogs Indoor House Open Toe Slippers

**Original Product Description**

ComfyWear Plush Clogs Womens Open Toe Slippers are made with quality material and plush fabric. Their soft and cozy feel will give you the spa comfort all day in your own home. The bright color heart design and contrast solid color with a bow on top makes them pretty and stylish to wear. These lightweight open toe slippers are fluffy and breathable and have slip resistant soles giving you a grip on all surfaces. Give the gift of coziness to your loved ones for all occasions!

**Textual User Profile**

- The user has plantar fasciitis and can only wear Clarks shoes.
- The user loves Clarks shoes and owns 4 pairs.
- The user is looking for comfortable shoes that they can wear all day long.
- The user is not interested in shoes from any other brand.
- The user is looking for boots that are just as comfortable as their Clarks flats.
- The user is not interested in boots that are not comfortable.
- The user is looking for a flattering fit.
- The user is not interested in clothes that make them feel exposed.
- The user loves the fabric and print of the clothes they buy.

**EAGLE Actions**

1. Use a high-density, supportive insole material known for its comfort and arch support, similar to what is used in Clarks shoes.
2. Redesign the open-toe clog into a closed-toe design with a more supportive, boot-like structure, while maintaining the plush comfort and lightweight feel.
3. Add a removable, washable insole for improved hygiene and easier cleaning.
4. Offer the slippers in a wider range of sizes to ensure a better fit for more customers.
5. Incorporate adjustable straps or closures to allow for a customized and secure fit.

**New Product Description**

ComfyWear Plush Closed-Toe Booties Womens Slippers are made with quality material and plush fabric, and feature a high-density, supportive insole known for its comfort and arch support, similar to what is used in Clarks shoes. Their soft and cozy feel will give you the spa comfort all day in your own home. The bright color heart design and contrast solid color with a bow on top makes them pretty and stylish to wear. These lightweight closed-toe booties offer a more supportive, boot-like structure while maintaining a plush and breathable feel and have slip-resistant soles giving you a grip on all surfaces. These booties also feature a removable, washable insole for improved hygiene and easier cleaning. These slippers are offered in a wider range of sizes to ensure a better fit for more customers. These booties incorporate adjustable straps for a customized and secure fit. Give the gift of coziness to your loved ones for all occasions!

## I.2 Nine West Women's Daydream Dress Pump

**Original Product Description**

Nine West offers a quick edit of the runways – pinpointing the must have looks of the season, and translating what is fun, hip, and of the moment. It is trend-right footwear that you will reach for in your closet again and again. Nine West is sure to be your trusted resource for everyday chic style.

**Textual User Profile**

- The user likes small, pretty, and subtle items.
- The user does not like items that look cheap.
- The user likes items that are flattering.
- The user does not like items that fit poorly.
- The user does not like items that are uncomfortable.
- The user does not like items that are a faded black color.
- The user likes items that are inexpensive.
- The user likes items that have pretty sides.
- The user does not like items that have a tightfitting neckline.

## EAGLE Actions

1. Revise the marketing materials to emphasize the delicate details and flattering silhouette of the Nine West Women's Daydream Dress Pump, highlighting its sophisticated and understated elegance instead of focusing solely on trendy aspects.
2. Add a subtle, decorative element to the side of the pump, such as a small, elegant cutout or a delicate embellishment, to enhance its visual appeal without overwhelming its understated elegance.
3. Offer the Nine West Women's Daydream Dress Pump in a rich, deep black color instead of faded black to enhance its luxurious appearance and appeal to customers who prefer a classic, sophisticated look.
4. Ensure the Nine West Women's Daydream Dress Pump is designed with a comfortable, non-restrictive fit to avoid discomfort and enhance the overall user experience.
5. Refine the marketing copy to emphasize the pump's affordability while maintaining its image of understated luxury, appealing to the user's preference for inexpensive yet high-quality items.

## New Product Description

Nine West presents the Daydream Dress Pump, a sophisticated and understatedly elegant addition to your wardrobe. Its delicate details and flattering silhouette, accented by a subtle, elegant side cutout, create a chic and timeless look you'll reach for again and again. Available in a rich, deep black, and designed for a comfortable, non-restrictive fit. Experience understated luxury without compromising on affordability.

## I.3    Lookin Fine Ladies Leggings

## Original Product Description

Brightly colored leggings are a huge fashion DO this year! Go casual and cool during the day, or dress them up for a girls night out in style. Comfortable, thin, and nearly invisible underneath clothes. You could wear these leggings underneath any skirt, tunic, or dress any day of the week, to almost any occasion! These durable, versatile, and fashionable leggings are almost yours, and quantities are limited! This style comes in many color options: Aqua, Black, Brown, Charcoal Grey, Fire Red, Hot Pink, Lime Green, Navy Blue, Neon Lime Green, Neon Orange, Neon Pink, Neon Yellow, Purple, Red, Turquoise, White and Yellow. Fabric is composed of 20fits most ladies. Hand wash in cold water and hang to dry, or for best results dry clean. If this is not the style of ladies leggings you were looking for today, then please continue browsing our other listings for additional styles and colors available.

## Textual User Profile

- The user likes products that are well-made and look expensive.
- The user is not afraid to spend money on quality products.
- The user is disappointed when products are cheaply made.
- The user is likely to return products that are not well-made.
- The user is likely to leave negative reviews for products that are not well-made.
- The user is likely to shop at stores that sell high-quality products.
- The user is likely to buy products that are made by well-known brands.
- The user is likely to read reviews before buying products.
- The user is likely to ask friends and family for recommendations before buying products.

## EAGLE Actions

1. Reformulate the leggings using a higher-quality, more luxurious fabric blend, such as a blend of sustainably sourced pima cotton and silk, while maintaining the same color options.
2. Add a higher-quality waistband with reinforced stitching for enhanced durability and comfort.
3. Offer the leggings in a wider range of sizes to better accommodate diverse body types.
4. Add a subtle, elegant logo tag to the leggings' waistband for a touch of branding and sophistication.
5. Offer a more sophisticated color palette, focusing on rich jewel tones and neutral earth tones, while still including a few classic bright options.

## New Product Description

Brightly colored leggings are a huge fashion DO this year! Go casual and cool during the day, or dress them up for a girls night out in style. Comfortable and luxurious, these leggings are nearly invisible underneath clothes. You could wear these leggings underneath any skirt, tunic, or dress any day of the week, to almost any occasion! These durable, versatile, and fashionable leggings are almost yours, and quantities are limited! This style comes in a sophisticated range of colors, including rich jewel tones, neutral earth tones, and a few classic bright options: Black, Brown, Charcoal Grey, Fire Red, Navy Blue, Purple, Red, Turquoise, Emerald Green, Sapphire Blue, and Rose Gold. Fabric is composed of a luxurious blend of sustainably sourced pima cotton and silk. Featuring a higher-quality waistband with reinforced stitching for enhanced durability and comfort, these leggings offer superior quality and lasting wear. Available in a wider range of sizes to accommodate diverse body types, these leggings ensure a perfect fit for every woman. Subtly embellished with an elegant logo tag on the waistband, these leggings exude understated luxury. Hand wash in cold water and hang to dry, or for best results dry clean. If this is not the style of ladies leggings you were looking for today, then please continue browsing our other listings for additional styles and colors available.

## I.4 Ridge Shirt Mens Packin Tee Concealment V-Neck Sleeveless Black 411BV

### Original Product Description

Mens Ridge Shirt: A Superior T-Shirt for Concealed Weapons! HOLSTER SOLD SEPARATELY. This Packin' Tee is designed for versatility, comfort and style. You can now have your gun at your side without the restricting discomfort from other types of holsters used in carrying concealed weapons. No longer is there a need to wear jackets of other heavy outerwear to conceal your weapon; lightweight shirts/clothing can be worn over the Packin' Tee the same as you would any other undershirt. HOLSTER SOLD SEPARATELY. 411W

### Textual User Profile

- The user is a parent who likes to buy clothes for their daughter.
- The user likes clothes that are adorable and look great.
- The user is concerned about the quality of the clothes and wants them to last.
- The user is willing to pay a higher price for better quality clothes.
- The user likes to buy clothes from Amazon.
- The user is less likely to buy clothes with negative reviews.

### EAGLE Actions

1. Rename the product to something more family-friendly and less suggestive of concealed weapons, such as "Ridge Shirt Men's V-Neck Undershirt."
2. Add a tagline emphasizing the shirt's softness and comfort, appealing to parents concerned about their child's comfort, such as "Incredibly soft and comfortable for all-day wear."
3. Highlight the durability and longevity of the shirt to appeal to parents seeking quality clothing, such as "Built to last, wash after wash."
4. Add details about available sizes and colors to cater to a wider range of children's clothing needs, such as "Available in a range of sizes and colors to perfectly match your child's wardrobe."
5. Specify that the undershirt is made from soft, high-quality cotton for enhanced comfort and durability.

### New Product Description

Men's Ridge Shirt V-Neck Undershirt: A superior undershirt designed for versatility, comfort, and style. This undershirt provides a smooth, comfortable fit under other clothing. Lightweight and breathable, it's perfect for everyday wear. Incredibly soft and comfortable for all-day wear. Built to last, wash after wash. Available in a range of sizes and colors to perfectly match your child's wardrobe. Made from soft, high-quality cotton for enhanced comfort and durability.

## I.5 Tahari Women's Spencer

### Original Product Description

Shearling lined mule. Tahari footwear is influenced by the fashion direction of the Elie Tahari designer apparel collections. Their footwear is designed to represent the colors and details that are important each season. They are known for their attention to detail and ornamentation and they work closely with the Elie Tahari design team to develop styles you will not see in other brands. They are a complete lifestyle brand offering weekend casuals, sandals, boots and career shoes for the women on the go, but at an affordable price.

### Textual User Profile

- This user prefers products that fit well.
- This user prefers products that are in their favorite color.
- This user does not like products that are uncomfortable.
- This user does not like products that are poorly made.
- This user does not like products that are expensive.
- This user prefers products that are made by a reputable brand.
- This user prefers products that have a lot of positive reviews.
- This user prefers products that are easy to use.
- This user prefers products that are durable.

### EAGLE Actions

1. Improve the sizing and fit of the Tahari Women's Spencer mule to ensure a comfortable and accurate fit for a wider range of foot sizes and shapes.
2. Add a description of the available colors and materials of the Tahari Women's Spencer mule to allow users to choose their preferred options.
3. Use more sustainable and ethically sourced materials in the production of the Tahari Women's Spencer mule, highlighting these efforts in the product description.
4. Add a cushioned insole to the Tahari Women's Spencer mule for enhanced comfort and support.
5. Highlight the positive customer reviews and testimonials for the Tahari Women's Spencer mule in the product description to build trust and credibility.

### New Product Description

Shearling lined mule. Tahari footwear is influenced by the fashion direction of the Elie Tahari designer apparel collections. Their footwear is designed to represent the colors and details that are important each season. They are known for their attention to detail and ornamentation and they work closely with the Elie Tahari design team to develop styles you will not see in other brands. They are a complete lifestyle brand offering weekend casuals, sandals, boots and career shoes for the women on the go, but at an affordable price. The Tahari Women's Spencer mule has been improved to offer superior sizing and fit, ensuring a comfortable and accurate fit for a wider range of foot sizes and shapes. The Tahari Women's Spencer mule is available in a range of colors and materials, including [list of colors and materials], allowing you to choose your perfect pair. The Tahari Women's Spencer mule is crafted using more sustainable and ethically sourced materials, reflecting our commitment to responsible manufacturing practices. For enhanced comfort and support, the Tahari Women's Spencer mule now features a cushioned insole. Discover why customers rave about the comfort and style of the Tahari Women's Spencer mule – read their reviews and testimonials here: [link to reviews].

## I.6  Cubavera Men's Short Sleeve Print and Ornate Embroidery Combination Panels Shirt

### Original Product Description

Cloak yourself in subtlety and refinement in the Cubavera ornamental print and embroidered combo panels shirt. This bright white number is actually a bit of a tease, because two vertical rows of stylish embroidery hint that the wearer is equally as comfortable with flamboyance as he is with understatement.

### Textual User Profile

- The user is looking for a high quality product.
- The user wants a product that is comfortable.
- The user is looking for a product that is not too expensive.
- The user is looking for a product that is easy to use.
- The user is looking for a product that is durable.
- The user is looking for a product that is safe.
- The user is looking for a product that is stylish.
- The user is looking for a product that is functional.
- The user is looking for a product that is unique.

### EAGLE Actions

1. Redesign the embroidery to feature a more understated, geometric pattern instead of the flamboyant style, maintaining the overall white color scheme.
2. Source the shirt fabric from sustainably produced, high-quality cotton, ensuring fair labor practices throughout the supply chain.
3. Add a reinforced, easily accessible pocket to the shirt for added functionality without compromising the sleek design.
4. Add a breathable, lightweight fabric lining to enhance comfort and breathability.
5. Offer the shirt in a wider range of subtly sophisticated colors, including navy, olive green, and charcoal grey, in addition to the bright white.

### New Product Description

Cloak yourself in subtle refinement with the Cubavera men's short sleeve shirt, crafted from sustainably produced, high-quality cotton, available in a range of sophisticated colors including bright white, navy, olive green, and charcoal grey. This shirt features two vertical rows of understated, geometric embroidery, offering a touch of unique style without overwhelming the minimalist design. The commitment to fair labor practices throughout the supply chain ensures a high-quality and ethically sourced product. A reinforced, easily accessible pocket adds convenient functionality without compromising the shirt's sleek aesthetic. A breathable, lightweight lining enhances comfort and breathability for all-day wear.

## I.7  Pack of 3 Pcs Printed Reversible Art Silk Sari Wrap Around Magic Skirts

### Original Product Description

Measure (Approx): Height: 36 inches Width: 47 inches. These skirts comes in assorted by stitching any two of various available random and assorted Indian saries fabric.

### Textual User Profile

- This user likes jewelry.
- This user is disappointed when receiving damaged items.
- This user cares about the aesthetics of products.
- This user cares about the safety of products.
- This user compares products from different vendors.
- This user chooses products based on quality and price.
- This user is willing to wait for a better product.
- This user is willing to pay more for a better product.
- This user is not afraid to give negative reviews.

## EAGLE Actions

1. Add individual product photos showing the exact print and color combinations of each skirt in the 3-piece pack, to manage customer expectations and reduce disappointment from receiving a random assortment.
2. Rename the product to highlight the unique, handcrafted nature of the skirts, such as "Pack of 3 Unique Handcrafted Reversible Art Silk Sari Wrap Skirts".
3. Add a detailed description of the sari fabrics used, specifying the type of silk and any embellishments, to emphasize quality and craftsmanship.
4. Highlight the versatility of the skirts by suggesting styling options and outfit combinations in the product description.
5. Specify the source of the sari fabrics and mention any ethical or sustainable sourcing practices, if applicable, to appeal to conscious consumers.

## New Product Description

Pack of 3 Unique Handcrafted Reversible Art Silk Sari Wrap Skirts. Measure (Approx): Height: 36 inches Width: 47 inches. These skirts come in assorted combinations of various Indian sari fabrics, sourced ethically from [Source Location/Supplier Name], with each 3-piece pack containing unique prints and color combinations shown in individual product photos. The skirts are crafted from high-quality art silk, with details varying from pack to pack, potentially including embellishments such as embroidery or zari work. Specific fabric types will be noted in individual product descriptions. Dress them up with jewelry and heels for an elegant evening look, or pair them with a simple top and sandals for a casual daytime style. The possibilities are endless!

