# OpenReview forum: "Embedding-Aligned Language Models"
_NeurIPS.cc/2024/Conference — NeurIPS 2024 poster_

### Official Review · Reviewer_gKnd · 2024-07-11

**Soundness:** 4
**Presentation:** 3
**Contribution:** 4
**Rating:** 7
**Confidence:** 4

**Summary:**

The paper proposes a method for prompting LLMs to generate content that optimizes an objective defined in a latent space through externally provided embedding spaces. To this end, they define a reinforcement learning agent (EAGLE) as follows: Given an entity (e.g. a movie description), an LLM is prompted to generate textual actions to change the movie in some way. Based on a chosen action, a separate LLM that acts as the environment performs the action (which is a textual prompt) on the given entity. By encoding the new entity in the embedding space, it's utility (externally given) can be computed. This RL agent is trained via a policy gradient method with a reference distribution, for which three alternatives are proposed. The results from the experimental evaluation with human raters on the MovieLens 25M dataset suggests that the RL agent significantly helps in finding novel movies (i.e., their descriptions) that users like.

**Strengths:**

* An interesting solution to an interesting problem with significant relevance to the NeurIPS community.
* To the best of my knowledge, the proposed solution is quite novel.
* A rigorous evaluation that shows that the proposed method helps in the narrow domain it was evaluated on.
* The paper is largely well-written, although some details remain unclear despite careful reading.

**Weaknesses:**

* The paper could improve on clarity in some parts. For example, the purpose of the reference distribution in this specific model, and the intuition behind the G-optimal design are not well explained (see questions below).
* The method is evaluated on only one use case and hence the evaluation is quite limited. It is unclear whether this method is good for other use cases as well, and how much case-specific tuning is needed, potentially taking away from the generality of the method.

**Questions:**

* The purpose of the reference policy is not clear to me. Why does the EAGLE agent need to be anchored to a reference policy at all? Why are the three proposed choices reasonable?
* The purpose of the G-optimal design is not clear. Intuitively, what does it achieve?
* Line 179 states that each x \in \mathcal{X} is associated with a set of actions by prompting an LLM for it, which would result in |A| * K actions (with possible duplicates). But in line 205 it is stated that only K actions are generated. Which one is it? If it's the former, is the action set dependent on x? Is a separate EAGLE trained for each x? If it's the latter, how do you obtain it?
* Line 254 states that the reference policy is trained like next token prediction with actions (or rather action texts)? as targets. Doesn't that mean it can (and will) generate entirely new actions at inference time?
* Table 2: How would a baseline without any reference policy perform?
* Table 3: Using GPT-4 at test time improves considerably over using Gemini Ultra (according to human raters). What is your explanation for this? Could it be that GPT-4 simply generates better movies? Is it generally plausible that using a better LLM as environment will yield better results without retraining (as opposed to merely being robust as stated in line 307)? How could this be tested?
* Although your method seems novel, I think your related work section doesn't discuss other methods that try to optimize prompts for LLMs using RL, e.g. [1]. Could you elaborate on how your method differs from them?

[1] https://aclanthology.org/2022.emnlp-main.222.pdf

**Limitations:**

Yes.

---

> ### Author Rebuttal · Authors · 2024-08-05
>
> Thank you for your positive review and helpful feedback. We appreciate the fact that you found our work interesting and novel. Please find our response to your comments and suggestions below.
>
> Re clarity:\
> We’ll elaborate further on the reference policy and its use. We’ll also move more information from the appendix in the main body of the paper to explain the intuition about the G-Optimal Design, in particular, Figure 2 in the appendix, and its explanation. Additionally, we’ll elaborate more on ELM, and reduce the mathematical burden in reading the paper.
>
> Re Experiments:\
> Thank you for pointing this out. We will add additional experiments using the public Amazon dataset, using user profiles and embeddings as defined in [1,2]. We will use this dataset to identify content gaps (i.e., new products) in product space. We believe the addition of these experiments will strengthen our work.
>
> Reference Policy:\
> A reference policy is commonly used in Policy Gradient algorithms with LLMs (see [1,3,4]). The reference policy is usually trained without RL to be a good anchor to the RL training procedure. That is, the reference policy ensures the RL algorithm does not diverge too much from an initial distribution that we deem to be good enough. It has been shown that removing the reference policy in RL with LLMs can hurt performance.\
> Unlike previous work, in our setting the use of the reference policy has an additional, key benefit. [5] shows that an epsilon-greedy RL strategy which uses a G-Optimal design distribution achieves efficient regret guarantees. In other words, w.p. epsilon, the algorithm samples an action from the G-Optimal design distribution. This induces favorable exploration. We therefore use a regularization to such a distribution in EAGLE, and show it indeed improves overall performance.
>
> As you point out, we choose three distributions for reference policies. The uniform distribution mimics a uniform exploration strategy that a naive epsilon-greedy algorithm would use. The best-next-state action distribution mimics the methodology that is used by current LLM-RL algorithms, which anchor the RL algorithm to a “good” policy (see e.g., [1,3,4]). Finally, G-Optimal design mimics the exploration strategy proposed in [5], which allows for efficient exploration, and improved overall performance.
>
> Further Intuition on G-Optimal Design:\
> We will add further intuition of the G-Optimal design to the main part of the paper (currently illustrated in Appendix C and Fig. 2). A core challenge in our framework stems from the potential bias introduced by using LLMs to generate actions, leading to an action space that favors specific directions within the embedding space (as visualized in Fig. 2 in the paper). This bias can significantly affect EAGLE's exploration and prevent it from discovering optimal novel entities. To mitigate this, we employ G-optimal design. By minimizing a form of worst-case variance (Definition 1), G-optimal design selects a diverse subset of k actions that maximize exploration, ensuring that no single direction in the embedding space is overly favored.
>
> State Dependent Action Space:\
> As you correctly point out, the action space generated is dependent on the state space. We generate K actions for every x. Indeed, actions are dependent on the entity x (e.g., a change to a movie may be significantly affected by its plot). We do not need to train a separate EAGLE model for each x, since x is given to the policy as input (i.e., we treat it as the description of the state). In our experiments the description includes a movie’s plot, reasons to like, dislike, and a user profile.
>
> Next Token Prediction:\
> You are correct. As we train our policy using next token prediction, it is not necessarily constrained by the set of candidate actions. The policy learns to generalize and generates different actions. This is also evident when we test EAGLE on movies it was not trained on, where it generates new actions.
>
> Re GPT-4:\
> We found that raters prefer results generated by GPT-4. This could generally be due to rater’s preferred generations of a GPT-4 environment vs. a Gemini-Ultra environment (e.g., creativity).
>
> Re Related Work:\
> Our work’s novelty focuses on two key aspects: (1) aligning to an embedding space, and (2) exploration using G-Optimal design. We are not aware of work using LLMs that has attempted to solve either of these specific problems. But apart from related work discussed, your suggestions are greatly appreciated. We will add further discussion of related work to our paper, including:
> - Aligning LLMs to other forms of modalities, such as graphs [6,7]
> - General exploration strategies in RL (irrespective of LLMs) [5,8,9]
> - Injecting embeddings in LLMs (not necessarily for alignment) [1,2,10]
>
> We expect these additions will help improve the clarity of the paper.
>
> **References:** \
> [1] Jeong, Jihwan, et al. "Factual and Personalized Recommendations using Language Models and Reinforcement Learning."  2023\
> [2] Tennenholtz, Guy, et al. "Demystifying embedding spaces using large language models." 2023.\
> [3] Roit, Paul, et al. "Factually Consistent Summarization via Reinforcement Learning with Textual Entailment Feedback." 2023.\
> [4] Ziegler, Daniel M., et al. "Fine-tuning language models from human preferences." 2019.\
> [5] Zhu, Yinglun, et al. "Contextual bandits with large action spaces: Made practical."2022.\
> [6] Tang, Jiabin, et al. "Graphgpt: Graph instruction tuning for large language models." 2024.\
> [7] Zhang, Mengmei, et al. "GraphTranslator: Aligning Graph Model to Large Language Model for Open-ended Tasks."  2024.\
> [8] Pathak, Deepak, et al. "Curiosity-driven exploration by self-supervised prediction." International conference on machine 2017.\
> [9] Zhou, Dongruo, Lihong Li, et al. "Neural contextual bandits with ucb-based exploration." 2020\
> [10] Cao, Xinlei, et al. "Injecting user identity into pretrained language models for document-level sentiment classification." 2022

---

> > ### Comment · Reviewer_gKnd · 2024-08-13
> >
> > Thank you for your responses. Most aspects are clearer to me now, I'd appreciate to find it in the next version of the paper. My evaluation remains unchanged.

---

> > > ### Author Response · Authors · 2024-08-13
> > >
> > > We appreciate your helpful comments which will improve the quality of our paper.

---

### Official Review · Reviewer_9y2i · 2024-07-20

**Soundness:** 3
**Presentation:** 4
**Contribution:** 3
**Rating:** 7
**Confidence:** 3

**Summary:**

This paper presents a method to steer an LLM’s generation towards optimal regions of a latent embedding space using reinforcement learning. The technique involves a language model to guide an LLM by modifying the textual representation of an entity. This work builds off previous work on embedding language models (ELM).

**Strengths:**

This paper presents a novel technique and shows promising results. In particular, EAGLE represents a strong improvement over ELM, which is a recent proposed technique.

**Weaknesses:**

* The experiments seem somewhat minimal

* The presentation is overly mathematical, and I found the appendix to be at times more informative than the main text. The mathematical language sometimes makes it more difficult to understand what experiments were actually done.

* The paper assumes knowledge of ELM (I had to read the ELM paper again in order to understand the details of EAGLE). The authors could make this slightly easier for the reader.

**Questions:**

How should we interpret the result that the Distance Score is higher for ELM than for EAGLE in Table 2?

**Limitations:**

* Given the nature of the work, it seems that this will be difficult to reproduce

---

> ### Author Rebuttal · Authors · 2024-08-05
>
> Thank you for your review and your helpful comments. We appreciate you finding our paper novel and a strong improvement over ELM. Please find our response to your suggestions and comments below.
>
> - We will add additional experiments using the public Amazon dataset, using user profiles and embeddings as defined in [1,2]. We will use this dataset to identify content gaps (i.e., new products) in product space. We believe the addition of these experiments will strengthen our work.
> - We will improve the writing in the paper to (1) reduce the mathematical burden, and (2) add additional explanation regarding ELM to help the reader understand this paper without having to thoroughly read that work.
> - As you correctly point out, ELM manages to achieve higher distance scores from the data. This result highlights the challenge of using EAGLE to move in embedding space, as it can only be controlled using  language. Indeed, ELM can better control its movement in embedding space, potentially allowing one to move outside the existing corpus. However, this comes with several disadvantages, which we discuss thoroughly in Appendix B, including: realizability of the decoded output; non-linearity of the embedding manifold (i.e., it is unclear how to move on it, and which  makes it hard to generalize too far from support); and the fact that the latent embedding may not encode enough semantic information for high-quality decoding. These challenges greatly reduce the ability of ELM to produce high-quality results, though it does manage to attain better distance scores.
>
> **References:** \
> [1] Jeong, Jihwan, et al. "Factual and Personalized Recommendations using Language Models and Reinforcement Learning." arXiv preprint arXiv:2310.06176 (2023). \
> [2] Tennenholtz, Guy, et al. "Demystifying embedding spaces using large language models." arXiv preprint arXiv:2310.04475 (2023).

---

### Official Review · Reviewer_aThJ · 2024-07-20

**Soundness:** 2
**Presentation:** 3
**Contribution:** 2
**Rating:** 5
**Confidence:** 4

**Summary:**

This work proposes training language models so that they follow objectives or utility functions which are defined in the embedding space. They define it as a reinforcement learning problem so that the EAGLE agent uses an actions prompt to probe the environment which is an LLM. The changed entity is embedded into the latent space where a reward is provided by the utility function. The experiments demonstrate better peformance compared to ELM. Further analysis shows interesting properties of environment transfer where the training and inference environments are different.

**Strengths:**

1) This method is interesting specially if combined with different kinds of utility functions which align with human preferences.

2) The method demonstrates strong experimental results albeit on a single dataset. The quality of the generation looks good.

**Weaknesses:**

1) The method relies on extensive prompt design specially for user profiles. How would this generalize to a new task. Can the prompt generation process be automated ?

2) The discussion on the computational complexity is not precise. The authors can pick a couple of scenarios and compare the computational complexity of ELM vs Eagle.

3) Experiments are conducted on a single dataset.

**Questions:**

Are the EAGLE generations in the appendix cherry picked or randomly selected ?

Did the authors experiment with a less capable LLM. Is the quality of the generation just due to the LLM ?

Was any other utility function considered ?

**Limitations:**

Yes.

---

> ### Author Rebuttal · Authors · 2024-08-05
>
> Thank you for your review and your helpful comments. Please find our response to your comments and suggestions below.
>
> Re Prompt Generation:
> 1. Much recent work (see, e.g., [1,2,3,4]) has shown the importance of designing task-specific prompts for solving tasks. While our work aligns LLMs with embeddings, there is no reason not to also use such approaches to improve the quality of the task-specific results. Automating prompt generation is a very important and interesting problem, but is beyond the scope of our work (and can be viewed as orthogonal to our approach).
> 2. Personalization requires injecting user information into the LLM. This can be done using, say, user embeddings, through fine-tuning (see e.g., [5,9]), or using user profiles (see, e.g., [1,5,6]). We use user profiles here, as our method directly uses an environment LLM without any fine tuning, a benefit of our approach.
> 3. Much work has recently focused on the generation of large datasets of user profiles and personas, intended to be used in personalization. This challenge is evolving, independent of our work, and will benefit in reducing the difficulty of generating such profiles. See, e.g., [5,6,7,8].
>
> Further Experiments: Thank you for pointing this out. We will add additional experiments using the public Amazon dataset, using user profiles and embeddings as defined in [5,10]. We will use this dataset to identify content gaps (i.e., new products) in product space. We believe the addition of these experiments will strengthen our work.
>
> Questions:
>
> - The results we show were not cherry picked, though we selected movies that are reasonably well-known.
> - We conducted experiments using less capable LLMs. In particular, our agent LLM is Gemini Nano (a small model, with just over 2B parameters). Our environment LLM is Gemini Pro, whereas evaluation is conducted on Gemini Ultra (see table 3 in paper).  Our results suggest that Gemini Pro training environment suffices to obtain high-quality inference results, when compared to training with Gemini Ultra, showing it did not improve overall performance.
> - As you’ve suggested, we will add experiments using the Amazon dataset.
>
>
> **References:** \
> [1] Park, Joon Sung, et al. "Generative agents: Interactive simulacra of human behavior." Proceedings of the 36th annual acm symposium on user interface software and technology. 2023. \
> [2] Kojima, Takeshi, et al. "Large language models are zero-shot reasoners." Advances in neural information processing systems 35 (2022): 22199-22213. \
> [3] Wei, Jason, et al. "Chain-of-thought prompting elicits reasoning in large language models." Advances in neural information processing systems 35 (2022): 24824-24837. \
> [4] Chen, Wenhu, et al. "Program of thoughts prompting: Disentangling computation from reasoning for numerical reasoning tasks." arXiv preprint arXiv:2211.12588 (2022). \
> [5] Jeong, Jihwan, et al. "Factual and Personalized Recommendations using Language Models and Reinforcement Learning." arXiv preprint arXiv:2310.06176 (2023). \
> [6] Chan, Xin, et al. "Scaling Synthetic Data Creation with 1,000,000,000 Personas." arXiv preprint arXiv:2406.20094 (2024). \
> [7] Shapira, Eilam, et al. "Can Large Language Models Replace Economic Choice Prediction Labs?." arXiv preprint arXiv:2401.17435 (2024). \
> [8] Wu, Bin, et al. "Understanding the Role of User Profile in the Personalization of Large Language Models." arXiv preprint arXiv:2406.17803 (2024). \
> [9] Cao, Xinlei, Jinyang Yu, and Yan Zhuang. "Injecting user identity into pretrained language models for document-level sentiment classification." IEEE Access 10 (2022): 30157-30167.\
> [10] Tennenholtz, Guy, et al. "Demystifying embedding spaces using large language models." arXiv preprint arXiv:2310.04475 (2023).

---

> > ### Comment · Reviewer_aThJ · 2024-08-14
> >
> > Thank you for your response. I think the additional experiment will be important to demonstrate the potential of this work. I have no further questions. I will keep my original evaluation.

---

### Official Review · Reviewer_3fAz · 2024-07-24

**Soundness:** 3
**Presentation:** 3
**Contribution:** 2
**Rating:** 5
**Confidence:** 4

**Summary:**

This paper proposes an algorithm to train an LLM-based agent *EAGLE*, that can align itself with existing domain-specific *latent embedding spaces* (e.g. embedding vectors in recommender systems, personalized advertising, and content creation) to discover novel content gaps and recommend new entities. It defines the problem setup as finding the optimal entity that maximizes the objective *utility function* $U(z; D)$, which is defined as the sum of user's and creator's utility, and the distance from all existing entities.

It formulates the setup as a Reinforcement learning optimization problem, in which MDP's state space consists of all possible entities, action space is defined by LLM-generated language prompts personalized to each user, the transition function is change in entity space obtained from the environment LLM, and the reward function is the *utility function* at horizon *H* and 0 otherwise. It considers several reference policies such as 1. Uniform 2. Greedy next step 3. *G-optimal* to constrain and regularize the learning objective. The *EAGLE* algorithm hence consists of first generating the candidate actions, training the reference policy, and then training the final policy given reference policy, pre-trained environment LLM, encoder and reward function.

The empirical experiments are performed on the dataset **Movielens-25M**, and the latent embeddings are the behavioral embeddings obtained via *matrix factorization* using *alternative least squares* trained on the user ratings. 100 candidate actions (50 generic, 50 personalized) are generated by prompting focusing on diverse aspects such as plot changes, character enhancements, and storyline. The quality is evaluated using a pool of human annotators who score the utility w.r.t to the Movielens user profile and the rater's user profile and the distance of generated movie from its anchor. The final metric used for evaluation is defined as the fraction of raters who preferred the generated movie to the original for each of the defined quality metrics. The proposed method is compared against the baseline method ELM (Embedding language model). It is observed that EAGLE outperforms ELM significantly on user and rater utility, but slightly decreases distance score. It also seems to be less sensitive to environment LLM used and improves scores primarily for poorly rated anchor movies, with a small decrease in perfectly rated movies. G-optimal design also seems to be more helpful for EAGLE compared to the reference policy.

**Strengths:**

- The novel formulation efficiently incorporates existing domain-specific latent embeddings and leverages the generative capabilites of the pretrained LLMs to surface content gaps. This can be very helpful in providing text-based personalized recommendations to users in real-world recommender systems.
- It is computationally cheaper and relatively more data-efficient to train compared to previous methods as it doesn't require learning an explicit decoder.

**Weaknesses:**

Some of the key design choices made by the proposed algorithm raise concerns on generalizibility to other real-world systems.

- Generating the action space requires significant prompt engineering effort with detailed criteria and in-context examples for each entity and the action set needs to be personalized to each user, and having personalized actions seems to be critical to the performance of EAGLE (Table 4). This raises serious concerns since the quality of the recommendation can be heavily biased towards the subset of the subjective criteria provided in the prompt, which may not always be possible to exhaustively define for each entity.
- The coverage of the latent embedding space would also be severely bottlenecked by the sampled candidate actions which may only explore a tiny portion of the latent embedding space in practice. It isn't clear how a practitioner would know if the action set criteria is diverse enough and how large of a hyperparameter *K* would result in a coverage that is sufficient.

While some of these issues are discussed in the limitation section, the experiments fail to provide a realistic picture to the algorithm since the proposed algorithm has far more domain-specific information compared to the baseline through the language prompt and in-context examples. One way to possibly address that may be reporting the efficacy of the algorithm with a simplified chosen language prompt in a zero-shot setting and stripping away any stylistic recommendations from the domain-specific information provided in the personalized actions prompt. While it would understandably perform worse in that setting, it would help demonstrate what portion of the performance boost is coming from the algorithm's exploration vs the initial user-specified selections, which isn't possible to conclude from the experiments in the paper.

**Questions:**

1. Is it possible to quantitatively estimate the coverage of the latent ambient space given the generated candidate action space? If not, could you give a description on how you would recommend a practitioner to iterate on the prompts and hyperparameters. Qualitatively comment on the difficulty of the process and any assumptions that you made along the way.

**Limitations:**

- The authors have addressed the limitations of their algorithm and the broader societal impacts.
- It could be further improved by discussing the summary of the practicality and generalizability aspects of the different components, as described in the above sections, in a separate paragraph of the Limitation section.

---

> ### Author Rebuttal · Authors · 2024-08-05
>
> Thank you for your thorough review and helpful feedback. We appreciate your positive assessment of our formulation and its strengths in surfacing content gaps using latent embeddings and the generative capabilities of LLMs. We address your concerns and questions below.
>
> As you correctly point out, our method relies on a design of an action space. We emphasize several key points:
> 1. While our experiments demonstrate the benefit of personalized actions, EAGLE is agnostic to the use of personalized actions. Our framework aligns an LLM’s generation to an embedding space w.r.t. some predefined criterion. While we demonstrate this through personalized generation, other objectives can be formulated that do not require specific user personalization. We’ll demonstrate this by adding an experiment not involving personalization to the paper.
> 2. You point out an important and fundamental problem that is often overlooked in RL -- designing an efficient, useful, and sufficiently exploratory action space is crucial for any RL problem, particularly those involving language, where the action space is combinatorially large. Contemporary RL methods using LLMs implicitly define an action space induced by a fine-tuned SFT model [1,2,3]. This is similar to our choice of the best next-state action reference policy. Nevertheless, the feasible set of actions must be defined, regardless. This set can be defined using a dataset of predetermined examples (e.g., demonstrations of creative generation), or, synthetic generation of data. Notably, our method is agnostic to how the candidate set of actions is generated.
> We are not aware of any available datasets for the task of creative generation, and therefore use synthetic generation of candidate actions.
> 3. The complete set of feasible actions is theoretically the set of all the possible utterances in the English language. This set of actions is not only too big, but also biased in terms of random exploration (which most RL algorithms use). To mitigate this bias, we leverage G-optimal design, which improves coverage within any given action set. That said, we acknowledge that this choice is a fundamental challenge for any problem involving RL, and particularly, RL with LLMs.
> 4. We present ELM as an alternative to showcase the trade-off between action space design and leveraging the expressive power of an environment LLM vs. directly decoding the embedding space (using ELM). While ELM avoids explicit action creation, it suffers from limitations in generalization and "out-of-manifold" issues. Our results highlight this trade-off, which we believe is a valuable contribution to the community. We discuss this trade-off and limitations thoroughly in Appendix B.
> 5. We fully agree that the limitations you raise in your review are valid. These arise in any approach involving RL with large action spaces, and particularly in language domains. One of our goals in this work is to emphasize these points to the research community, which explicitly highlights these challenges and offers potential solutions, such as G-optimal design, as a starting point for future research. As such we view these points as strengths of our paper.
>
> Regarding Coverage: You correctly identify the challenge of estimating coverage of the latent embedding space. As you mention, we cannot exhaustively explore all possible actions.  However:
> 1. G-optimal Design provides an exploration metric. Specifically, it provides a quantitative measure of how exploratory the action space is.
> 2. In our work we use a G-Optimal design that is constrained to the set of uniform policies over subsets of the action set. Nevertheless, one may learn a more general G-Optimal design (as defined by Definition 1, and studied in [4]), where an arbitrary distribution over the action set is learned. We did not find that this added  complexity is needed in practice. Additionally, for selecting the number of actions K, one may use the exploration approximation constant C in Definition 1 to select an optimal value for K (which achieves highest coverage).
> 3. A fundamental challenge remains when using an LLM environment:  even with G-optimal design, achieving complete coverage of the embedding space might be hard or even impossible. This is because the environment LLM's capabilities ultimately constrain the states we can explore in the embedding space. In other words, there may not exist actions that would move us in certain directions in the embedding space. This limitation is inherent to using LLMs, an important point we raise in the paper. We will emphasize this further.
> 4. Most LLM work relies solely on SFT (our variant of best next next-state action reference policy), implicitly limiting exploration to the SFT model's capabilities.  We believe our explicit action space design and use of G-optimal design is a valuable step towards more robust and diverse exploration in LLMs.
>
> We note that Appendix B emphasizes many of the limitations and tradeoffs of our algorithmic approach. We believe these key trade-offs should be viewed as strengths of our work, rather than weaknesses, as they raise fundamental challenges that, to date, have been overlooked by the research community, while providing a novel method for aligning language models to embedding spaces.
>
> **References:**\
> [1] Ziegler, Daniel M., et al. "Fine-tuning language models from human preferences." arXiv preprint arXiv:1909.08593 (2019). \
> [2] Roit, Paul, et al. "Factually Consistent Summarization via Reinforcement Learning with Textual Entailment Feedback." Proceedings of the 61st Annual Meeting of the Association for Computational Linguistics (Volume 1: Long Papers). 2023. \
> [3] Jeong, Jihwan, et al. "Factual and Personalized Recommendations using Language Models and Reinforcement Learning." arXiv preprint arXiv:2310.06176 (2023). \
> [4] Zhu, Yinglun, et al. "Contextual bandits with large action spaces: Made practical." International Conference on Machine Learning. PMLR, 2022.

---

> > ### Comment · Reviewer_3fAz · 2024-08-12
> > **Updated review scores**
> >
> > Thanks for the clarifications on the concerns and questions. I have updated the review score to (5: Borderline accept) for the following reasons:
> > 1. The authors promised to add a new experiment to demonstrate if EAGLE is agnostic to personalized actions. I have kept the score as borderline as while the analysis would be helpful regardless, the impact of personalization on performance is unclear at this point.
> > 2. I am satisfied with their response regarding the limitations I mentioned being a fundamental challenge in any RL application with LLM and the importance of studying them anyway. But I'm still not convinced of the motivation behind G-optimal design specifically from the description and review responses and addressing the practicality concerns I raised in my review. I strongly encourage clarifying those two points as It would improve the quality of the paper further and is something easily possible to do in the final draft.

---

> > > ### Author Response · Authors · 2024-08-12
> > >
> > > We appreciate your response and updating your review score. Your suggestions will help improve the quality of our paper. Following up we are updating the paper to (1) include an additional experiment on the Amazon Public dataset, and (2) clarify the use of G-Optimal design more exhaustively. Particularly for (2), we will move more explanation from Appendix B and C to the main paper and further base the use of G-Optimal design on [4].

---

### Decision · Program_Chairs · 2024-09-25

**Decision:**

Accept (poster)

**Comment:**

This paper presents EAGLE, an algorithm that leverages RL for training LLMs to discover novel content gaps and recommend new entities with existing domain-specific latent embedding spaces. They formulate the problem as markov decision process and utilize reference policies to regularize the learning objective. They demonstrates improvements over the baseline method ELM in terms of user and rater utility on the Movielens-25M dataset and show evidence as an effective content discovery and recommendation framework.

The paper presents a novel solution to a relevant problem and demonstrate its effectiveness with a rigorous evaluation in a specific domain. Although some details on the experimentation and method was missing in the original submission, the authors provided additional evidence and some reviewers were happy about it. Authors also agreed to add more information in the final submission (e.g, about personalization, related work, and others)

One challenged raised by reviewers was about generalizability of EAGLE due to its reliance on personalized actions and the need for significant prompt engineering effort (which may lead to biased recommendations and limited exploration of the latent embedding space).
The authors argued that EAGLE is agnostic to personalized actions and can be formulated with other objectives that don't require user personalization. Authors agreed to add an experiment to demonstrate EAGLE's performance without personalization.